# Pause Tokens Strictly Increase the Expressivity of Constant-Depth Transformers

**Charles London**
Department of Computer Science
University of Oxford
Oxford, OX1 3QG
charles.london@cs.ox.ac.uk

**Varun Kanade**
Department of Computer Science
University of Oxford
Oxford, OX1 3QG
varun.kanade@cs.ox.ac.uk

## Abstract

Pause tokens, simple filler symbols such as "...", consistently improve Transformer performance on both language and mathematical tasks, yet their theoretical effect remains unexplained. We provide the first formal separation result, proving that adding pause tokens to constant-depth, logarithmic-width Transformers strictly increases their computational expressivity. With bounded-precision activations, Transformers without pause tokens compute only a strict subset of $AC^0$ functions, while adding a polynomial number of pause tokens allows them to express the entire class. For logarithmic-precision Transformers, we show that adding pause tokens achieves expressivity equivalent to $TC^0$, matching known upper bounds. Empirically, we demonstrate that two-layer causally masked Transformers can learn parity when supplied with pause tokens, a function that they appear unable to learn without them. Our results provide a rigorous theoretical explanation for prior empirical findings, clarify how pause tokens interact with width, depth, and numeric precision, and position them as a distinct mechanism, complementary to chain-of-thought prompting, for enhancing Transformer reasoning.

## 1 Introduction

Transformers [38] have become ubiquitous in modern machine learning, achieving state-of-the-art performance across a wide range of tasks, particularly in natural language processing. Despite their empirical success, many aspects of their computational behaviour remain poorly understood. An intriguing recently observed phenomenon is that adding pause tokens (e.g. a "..." token) to Transformer inputs can improve performance on some question-answering and mathematical tasks [30, 14]. This phenomenon has raised questions about the computational role of such seemingly meaningless tokens, particularly in the context of chain-of-thought prompting and the faithfulness of intermediate reasoning steps.

In this paper, we provide a theoretical explanation for how pause tokens might improve Transformer performance. Specifically, we analyse how they affect the expressivity of Transformers through the lens of circuit complexity. Our contributions are as follows:

1. **Pause tokens allow Transformers to simulate Boolean circuit classes exactly.** We prove that constant-precision Transformers with a polynomial number of pause tokens are equivalent in expressivity to $AC^0$ (constant depth circuits with and/or/not gates), while logarithmic precision Transformers with pause tokens match $TC^0$ (circuits like $AC^0$ with threshold gates).

2. **Pause tokens strictly increase expressivity in constant-precision Transformers.** We prove a separation: Transformers without pause tokens are fundamentally less expressive than those with.

39th Conference on Neural Information Processing Systems (NeurIPS 2025).

3. **Empirically, pause tokens help causally masked Transformers learn functions requiring global computation.** We show that two-layer causally masked Transformers struggle to learn parity (which belongs to $\mathsf{TC}^0$) unless pause tokens are introduced. This provides some insight into how pause tokens interact with architectural constraints in practice.

We consider both uniform and non-uniform Transformer and circuit classes (see Definition 3.6), covering the cases where Transformer parameters and positional encodings can be entirely arbitrary, and where they must be generated by an efficiently computable procedure.

Our results provide a principled explanation for recent empirical findings on pause tokens while also highlighting broader implications for Transformer expressivity. In particular, they suggest that a Transformer's computational power is not solely determined by depth and width, but also by how computation is distributed across the token sequence. The addition of pause tokens effectively expands the available computational workspace of the model, allowing it to implement more expressive computations within the same architectural constraints. These results have implications for the faithfulness of chain-of-thought reasoning, the tradeoffs between width and depth in Transformer architectures, and the impact of quantization on Transformer computation.

The remainder of this paper is organised as follows. Section 2 reviews prior work on Transformer expressivity and the use of pause tokens (more related work is in the appendix). Section 3 introduces key concepts from circuit complexity and formally defines the Transformer models we study. Section 4 presents our main theoretical results, establishing the expressivity of Transformers with and without pause tokens. Section 5 provides empirical evidence that pause tokens facilitate learning of parity under causal masking. Finally, Section 6 discusses implications and Section 7 poses open questions.

## 2 Related Work

We first note the difference between the use of pause tokens and chain-of-thought (CoT, Wei et al. [39]) in Transformer computation. While both methods purport to increase the computational space or memory of the Transformer to enable it to solve more problems, they do so in very different ways. In CoT, a Transformer generates tokens autoregressively and attends to its own output, enabling deeper sequential computation. With pause tokens, we simply pad the input to the Transformer with filler tokens, enabling "wider" parallel computation. While pause tokens are more efficient (as they can be computed in parallel), they are strictly less expressive than CoT [28, 22].

**Transformers with pause tokens**: The notion of adding padding to a Transformers input to improve performance traces back to Burtsev et al. [5], in which they prepended memory tokens to the input to a Transformer encoder, but achieved only marginal gains. Lanham et al. [21] experimented with replacing Transformer CoT tokens with blank "..." tokens at test time, and found they decreased performance versus CoT. The first work to empirically demonstrate an advantage in using pause tokens was Goyal et al. [14], in which they pre-trained a Transformer to use between 0 and 50 pause tokens before responding. This led to increased performance on natural language and mathematical benchmarks including CommonsenseQA [35] and GSM8K [10]. Since then, Pfau et al. [30] have replicated this improvement on simple mathematical tasks and proposed quantifier depth as the theoretical explanation.

**Formal models of Transformers**: Since the invention of the Transformer, there have been many attempts to formally characterise their computational power. Initial works determined that Transformers were Turing complete [31], but required infinite precision for the result. More recent works have used tools from theoretical computer science, including circuit complexity, communication complexity, and formal language and automata theory, to analyse more realistic models of the Transformer and derive limitations on the languages they can recognise and problems they can solve. For machine learning, language recognition can be thought of as binary classification of input strings. This section covers work using circuit complexity, but further related work can be found in Appendix A.

**Circuit complexity.** Hao et al. [16] showed that log-precision Transformers using unique hard attention (attending to only a single position) are contained within $\mathsf{AC}^0$, while averaging hard attention Transformers can express non-$\mathsf{AC}^0$ languages. Merrill and Sabharwal [27] proved that log precision Transformers (with standard softmax attention) are contained within logspace-uniform $\mathsf{TC}^0$, while Chiang [9] extended these results to show that Transformers with $O(\text{poly}(n))$ precision are in uniform $\mathsf{TC}^0$. Further works have also used the circuit complexity perspective to determine

the expressive power of both constant- and log-precision Transformers using intermediate chain-of-thought steps, showing that they are equivalent to P (uniformly, or P/poly non-uniformly) [22, 28].

## 3 Preliminaries

In this section, we introduce the relevant computational complexity concepts, including circuit classes and logspace uniformity, as well as definitions of the Transformer model we are considering and fixed precision arithmetic.

### 3.1 Boolean Circuits

Boolean circuits are a model of computation that uses logical operations to process binary inputs and produce binary outputs. They are a natural way to formalise parallel computations (like Transformers), as their depth corresponds roughly to the parallel time required to compute a function, and their width corresponds to resource requirements at each level of parallel computation.

**Definition 3.1.** (Boolean circuits). A Boolean circuit $C_n$ is a directed acyclic graph with $n$ sources and one sink, computing a function $\{0,1\}^n \to \{0,1\}$. Non-source vertices are gates labeled with logical operations. The output $C_n(x)$ is determined recursively. $|C_n|$ is the number of gates; depth is the longest path from any source to the sink.

We restrict ourselves to two classes: $AC^0$ and $TC^0$, with constant $O(1)$ depth and at most polynomial $poly(n)$ size. $AC^0$ circuits use NOT, AND, and OR gates with unbounded fan-in. $TC^0$ circuits use threshold gates $f(x) = I\left[\sum_{i=1}^{m} x_i > \theta\right]$ and $f(x) = I\left[\sum_{i=1}^{m} x_i < \theta\right]$. It is known $AC^0 \subsetneq TC^0$ [13]. $AC_\ell^0$ and $TC_\ell^0$ are these classes restricted to depth $\leq \ell$.

### 3.2 Circuit Families and Logspace Uniformity

**Definition 3.2.** (Circuit families). Let $T : \mathbb{N} \to \mathbb{N}$ be a function. A $T(n)$-size circuit family is a sequence $\{C_n\}_{n \in \mathbb{N}}$ of Boolean circuits, where $C_n$ has $n$ inputs and a single output, and $|C_n| \leq T(n)$ for all $n$.

Circuit families implicitly define a language that they accept, as follows:

**Definition 3.3.** (Language recognition). A circuit family $\{C_n\}$ recognises $L \subset \{0,1\}^*$ if, for all $x \in \{0,1\}^*$, $C_{|x|}(x) = 1$ if and only if $x \in L$.

Without further constraints, circuit families can recognise undecidable languages, such as variants of the halting problem. To avoid this, we use uniform circuits, which can be constructed by some computable function. A family $\{C_n\}$ is logspace uniform if there is a Turing machine using $O(\log n)$ space that outputs $C_n$ given $1^n$ (for a formal definition, see Definition B.2). Logspace uniformity ensures that circuits can be constructed systematically, avoiding non-uniform "hard-coded" solutions for each input size.

### 3.3 Limited Precision Arithmetic

We follow Li et al. [22] in using a fixed-point numerical representation, with 1 bit for the sign, $p(n)$ bits before the point, and $p(n)$ bits afterwards. When working with constant precision $p(n) = O(1)$, and for logarithmic precision $p(n) = O(\log n)$. We will use the shorthand $p = p(n)$.

**Definition 3.4** (Fixed precision arithmetic). For a fixed precision level $p$:

- Values are represented in $\mathbb{F}_p = \{c \cdot k \cdot 2^{-p} : c \in \{-1, 1\}, k \in \mathbb{N}, 0 \leq k \leq 2^{2p} - 1\}$.

- Arithmetic operations (e.g., addition, multiplication) round results to the nearest representable value in $\mathbb{F}_p$. We define by $[x]_p : \mathbb{R} \to \mathbb{F}_p$ the operation that rounds to the nearest representable number. When addition is iterated, we operate left to right, rounding after each operation (so it is no longer associative).

- Saturation arithmetic is used: values exceeding the representable range $[-B_p, B_p]$, where $B_p = 2^p - 2^{-p}$, are clamped to $\pm B_p$.

We justify these choices as a reasonable approximation of how arithmetic is done in quantized neural networks. These quantized networks often use a form of outlier clipping (saturation arithmetic) to avoid issues with overflow, and round numbers to the closest number in the representable range [8].

### 3.4 Transformer Definition

All theoretical results in this work apply to both decoder-only and encoder-only Transformers models (i.e. with and without causal masking). In our proofs on the lower bounds of Transformer expressivity, we assume causal masking as this setting is more restrictive, but results still hold if it is removed.

**Definition 3.5** (Transformer). A Transformer consists of $l$ layers, each composed of a multi-head masked self-attention mechanism and a feedforward network. Given an input sequence $X = (x_1, \ldots, x_n)$, where $x_i \in \mathbb{R}^d$:

- **Self-Attention:** Each head computes a weighted sum of a linear transform of the input tokens

$$X \leftarrow XW^V \cdot \mathrm{softmax}\left(XW^{QK}X^\top + M\right) \tag{1}$$

  where $W^{QK}, W^V \in \mathbb{R}^{d \times d}$ and $M \in \mathbb{R}^{n \times n}$. In the case that the Transformer is causally masked $M$ is lower triangular, with $-B_p$ above the diagonal, and all other elements zero. Without masking, $M$ is the zero matrix. The models we consider have a fixed number of attention heads.

- **Feedforward Network:** A position-wise feedforward network, with constant depth and $O(d)$ width, is applied to each token after each attention layer.

- **Positional Encodings:** Fixed or learned positional encodings $\phi(i, n) \in \mathbb{R}^d$ are added or appended to each input token $x_i$ to encode token positions.

- **Residual Connections:** Each sublayer (attention/feedforward) has residual connections.

We will define by $\mathsf{TF}[p(n), d(n), b(n)]$ the family of problems $\mathcal{L} : \{0, 1\}^n \to \{0, 1\}$ for which there is an integer $l$ such that there exists an $l$-layer Transformer with precision $2p(n)$ [1], embedding dimension $d(n)$, and $b(n)$ additional blank tokens that computes $\mathcal{L}(x)$ on any $x \in \{0, 1\}^n$.

In this work, we will largely operate with classes of Transformers with some range of functions $p(n), d(n), b(n)$. In the constant precision case, the classes we consider are:

1. $\mathsf{TF}[1, L, 0] \coloneqq \bigcup_{c \in \mathbb{N}} \mathsf{TF}[c, c \log n, 0]$,
2. $\mathsf{TF}[1, L, P] \coloneqq \bigcup_{c \in \mathbb{N}} \mathsf{TF}[c, c \log n, n^c]$,
3. $\mathsf{TF}[1, L, Q] \coloneqq \bigcup_{c \in \mathbb{N}} \mathsf{TF}[c, c \log n, 2^{(\log n)^c}]$.

These are Transformers with constant precision, logarithmic embedding size, and no, polynomial, and quasi-polynomial pause tokens, respectively. We choose logarithmic embedding size as we feel it is the most practically relevant to current architectures, while still enabling attention over the entire input sequence. To define the uniform class $\mathsf{TF}[1, L, Q]$, we also have to define a notion of polylogspace-uniformity, which can be defined analogously to logspace-uniformity (see Definition B.3).

In the logarithmic precision case, the classes $\mathsf{TF}[L, L, \cdot]$ are defined analogously, with $p(n) = c$ replaced by $p(n) = c \log n$. We denote by $\mathsf{TF}_l$ the family of Transformers with depth $\leq l$.

### 3.5 Logspace Uniform Transformers

Prior work on Transformer expressivity has largely focused on two extremes of uniformity. On the one hand, Li et al. [22] analyses non-uniform Transformers, where separate parameters can be freely defined for each input length $n$, effectively allowing different models for different input sizes. On the other hand, Merrill and Sabharwal [27] consider Transformers where the positional encoding function $\phi(i)$ is the same for all input lengths. If we want to allow the embedding size to increase for longer input sequences (necessary in the constant precision case), our embedding function must be

---

[1]$2p(n)$ as we have $p(n)$ bits before the fixed point and $p(n)$ bits after.

able to depend on $n$ i.e. $\phi(i, n)$. In practice, when a model with a larger context window is needed, often the embedding size is increased (and a whole new model is trained).[2]

We introduce a logspace-uniform Transformer class as a middle ground between these approaches. Our construction captures a more realistic computational model while still enforcing constraints that prevent edge cases where Transformer families could encode undecidable problems through arbitrary variations in their parameters.

**Definition 3.6** (Logspace uniform Transformer families). A family of Transformers $\{T_n\}$ is logspace uniform if there exists Turing machines $M_1$ and $M_2$ operating with logarithmic space such that: $M_1$ takes as input $1^n$ and outputs the weights and biases of $T_n$; $M_2$ takes as input $1^n$ and the binary encoding of an index $i$ and outputs $\phi(i, n)$.

This formulation allows for naturally size-dependent embeddings and weights but prevents freely designing entirely unrelated models for different $n$. This definition also allows us to draw natural connections with uniform circuit classes, ensuring that expressivity results reflect constraints on efficiently computable models rather than arbitrary, input-length-specific designs.

## 4 Expressiveness of Transformers with pause tokens

Given the empirical improvements observed with pause tokens, a natural question is whether these tokens simply aid optimisation, or fundamentally alter the Transformer's expressivity. In this section, we present our main theoretical contributions, which establish a rigorous computational foundation for increasing Transformer expressivity with pause tokens.

### 4.1 Constant precision Transformers and $\mathsf{AC}^0$

We begin with the case where the Transformer operates with constant-precision arithmetic. Understanding expressivity in this regime has become more important due to the widespread use of low-bit quantized LLMs. First, we show that the class of constant-precision Transformers with a polynomial number of pause tokens is equivalent to $\mathsf{AC}^0$. This demonstrates that the pause tokens can act as intermediate computational units, allowing a Transformer to effectively encode the computation of a Boolean circuit within its residual stream.

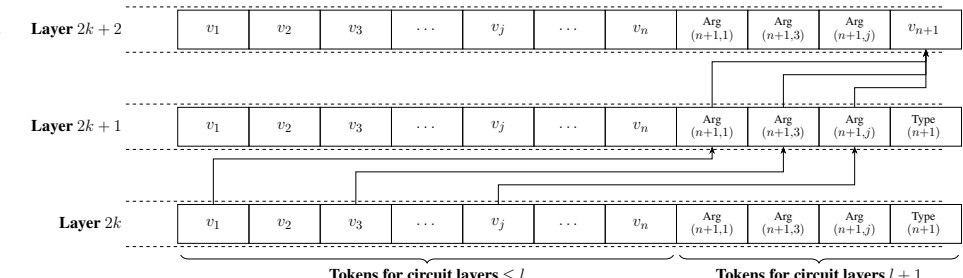

Figure 1: Two layers of a Transformer with pause tokens can simulate a layer of a Boolean circuit. In the first layer, inputs to the gates in the layer are copied to argument positions. In the second layer, these arguments are combined at the gate position to compute the output of the gate. $v_i$ represents the value of vertex $i$ in the circuit, whether that be an input or a gate. $\text{Arg}(i, j)$ tokens denote an edge from gate $j$ to gate $i$, and $\text{Type}(i)$ tokens denote a gate $i$. The red arrow represents the direction of computation.

Our first result demonstrates that $\mathsf{TF}[1, L, P]$ is computationally equivalent to $\mathsf{AC}^0$ (both uniformly and non-uniformly), even when the Transformer has causal masking.

**Theorem 4.1.** $\mathsf{TF}[1, L, P] = \mathsf{AC}^0$.

---

[2]For example, when moving from Llama2 to Llama3, Meta doubled the embedding dimension of the largest model and increased the context window from $4096$ to $128, 000$ tokens [36, 12].

To prove this equivalence, we show that (1) any function in $\mathsf{AC}^0$ can be computed by a Transformer in $\mathsf{TF}[1, L, P]$, and (2) every function computed by such a Transformer is in $\mathsf{AC}^0$. We sketch a proof below.

**Simulating $\mathsf{AC}^0$ with Transformers.** Let $C_n$ be a logspace-uniform $\mathsf{AC}^0$ circuit family computing a Boolean function $f : \{0, 1\}^n \to \{0, 1\}$. We construct a causally masked Transformer that simulates $C_n$ as follows:

- Token representation: The Transformer receives as input a sequence of length $n$ encoding a binary string $x$. A polynomial number of pause tokens are appended, each representing an edge or vertex in $C_n$.
- Positional encodings: The Transformer's positional encodings represent the structure of $C_n$, specifying the connectivity between input bits, intermediate gates, and the final output gate. Attention weights ensure that each gate token attends only to its input tokens.
- Computation by Transformer layers: Each pair of Transformer layers $2k + 1$ and $2k + 2$ mimic a single layer of the circuit $C_n$. The first attention layer copies the values of previously computed gates to the edge tokens, and the second attention layer computes the (rounded) sum of the inputs to each gate (see Figure 1). The feedforward layers are used as thresholds to compute the exact bit value of the gate output.

A full construction, including explicit positional encodings, weight matrices, and feedforward computations can be found in Theorem C.1.

**Transformers are contained in $\mathsf{AC}^0$.** Every function necessary to implement a Transformer in $\mathsf{TF}[1, L, P]$ can be implemented by an $\mathsf{AC}^0$ circuit. The positional encoding function is logspace-uniform, with $O(\log n)$ size outputs, and is thus computable in (uniform) $\mathsf{AC}^0$ [27]. Attention mechanisms and feedforward layers involve multiplication, division and exponentiation of a constant amount of constant precision numbers, and addition of at most a polynomial amount of constant precision numbers, all of which are in (uniform) $\mathsf{AC}^0$. Since a Transformer has a fixed number of layers, its overall computation is in $\mathsf{AC}^0$. See Theorem C.5 for the detailed proof.

Given this equivalence, we can move on to proving separations between Transformers with and without pause tokens. Our next result demonstrates that in the non-uniform case, a Transformer with a polynomial number of pause tokens is strictly more expressive than one without.

**Corollary 4.2.** *For the non-uniform class of Transformers,* $\mathsf{TF}[1, L, 0] \subsetneq \mathsf{TF}[1, L, P]$.

It follows from the result of Li et al. [22] that $\mathsf{TF}[1, L, 0] \subsetneq \mathsf{AC}^0$. Without pause tokens, the number of available computational units in the Transformer is limited by the sequence length $n$. We can obtain an upper bound on the size of the circuits it can simulate by considering the size of the circuits that can simulate it. This will be of some fixed polynomial size $O(n^k)$ (dominated by the size of the circuit needed to add $n$ numbers). By the non-uniform size hierarchy for Boolean circuits [1], there exist functions in non-uniform $\mathsf{AC}^0$ that require circuit size greater than $O(n^k)$. The class of Transformers without pause tokens is strictly less expressive than $\mathsf{AC}^0$, and therefore less expressive than $\mathsf{TF}[1, L, P]$.

In the uniform case, the separation is weaker, as there is no equivalent of the non-uniform size hierarchy. Instead, we make use of the *fixed-depth* uniform size hierarchy for $\mathsf{AC}^0$ (see Lemma D.3) to prove a separation for Transformers of some fixed depth $l$.

**Theorem 4.3.** *For the uniform class of Transformers,* $\mathsf{TF}_l[1, L, 0] \subsetneq \mathsf{TF}_l[1, L, Q]$.

In this case, our separation is between fixed-depth Transformers with no blank tokens, and those with a *quasi-polynomial* number of blank tokens (for the proof see Theorem D.4).

It should be clear that the addition of pause tokens does not enable the Transformer class to solve problems that require more serial computation steps, but instead increases the parallel width of computation. While it seems intuitively obvious that a deeper Transformer (still of fixed depth) that can perform more sequential steps is more expressive, it is not immediately obvious what is not computable at a fixed depth with a polynomial number of pause tokens. In the interest of rigour, we prove the following lemma.

**Lemma 4.4.** *For any depth $l$ there exists a constant $m > l$ such that* $\mathsf{TF}_l[1, L, P] \subsetneq \mathsf{TF}_m[1, L, P]$.

The proof of this lemma is based on the result of Hastad [17], who showed that there exist functions within $\mathsf{AC}^0_{l+1}$ that are not computable by any poly-size circuit in $\mathsf{AC}^0_l$.

## 4.2 Logarithmic Precision Transformers and $TC^0$

In the previous section, we established that constant precision Transformers with pause tokens are equivalent to $\mathsf{AC}^0$. We now extend this analysis to (uniform and non-uniform) logarithmic-precision Transformers, showing that they are equivalent to the strictly larger class $\mathsf{TC}^0$: constant-depth Boolean circuits with threshold gates.

**Theorem 4.5.** $\mathsf{TF}[L, L, P] = \mathsf{TC}^0$.

This result is equivalent to that obtained in Merrill and Sabharwal [27], that Transformers with polynomial advice (extra input to a TM that is allowed to depend on the input length $n$ but not the input itself) contain $\mathsf{TC}^0$, but our definition of Transformer uniformity leads to equality in the uniform case.

The proof structure is very similar to the constant precision case, with the key difference being that logarithmic precision allows attention layers to perform threshold-like computations, as required to simulate $\mathsf{TC}^0$. In the reverse direction, it is necessary to show that the iterated sum of a polynomial amount of logarithmic precision numbers is in $\mathsf{TC}^0$. For the full proof, see Appendix C.2.

While this result establishes a theoretical equivalence, we are unable to prove a strict separation between $\mathsf{TF}[L, L, 0]$ and $\mathsf{TF}[L, L, P]$ as we did in the constant precision case. Unlike $\mathsf{AC}^0$, where several superpolynomial bounds are well established, $\mathsf{TC}^0$ lower bounds are still an open area of research. The best known *wire* lower bounds for $\mathsf{TC}^0$ are of the form $n^{1+c^{-d}}$ for some $c > 1$, and a $\mathsf{TC}^0$ circuit simulating a Transformer trivially requires $\Omega(n^2)$ wires due to the attention between all tokens [6, 7].

## 5 Experiments: learning parity with pause tokens

**Parity as a $\mathsf{TC}^0$ benchmark.** Parity is a canonical problem within $\mathsf{TC}^0$, separating $\mathsf{AC}^0$ from $\mathsf{TC}^0$ by requiring threshold-like computations to count the number of set bits modulo 2 [13]. While we are unable to theoretically separate $\mathsf{TF}[L, L, 0]$ and $\mathsf{TF}[L, L, P]$, prior work has established that learning parity is difficult for Transformers without pause tokens [3, 15]. Our experiments investigate whether pause tokens enable practical learning of parity in a 2-layer Transformer.

We hypothesize that some of the observed performance increases afforded by pause tokens are due to the restrictive nature of causal masking in decoder-only Transformers. This masking limits the model's ability to aggregate information globally over the input sequence, as only the final token can attend to the entire sequence. For parity, a standard construction for a 2-layer threshold circuit involves $n$ threshold gates in the first layer, with each gate taking all $n$ input bits as input (i.e. global information, see Appendix E). A causally masked Transformer may struggle to simulate this, while pause tokens, by providing additional "gates" that can attend to the entire input, offer a mechanism to bypass this bottleneck.

Our main reason for focusing on the $\mathsf{TC}^0$ regime is practical. To effectively simulate the $\mathsf{AC}^0$ regime would require that our precision $< \log n$, requiring either scaling the input to unrealistic lengths (for our compute budget) with full precision or training heavily quantized models, which we found to be unstable and non-performant.

**Experimental setup.** We train a 2-layer, 4-head GPT-2-style Transformer [32] on bit sequences of length $\in \{20, \dots, 300\}$. The model is trained to compute the parity of the input sequence, where the final token's output is used for classification. To study the impact of pause tokens and causal masking, we consider the following scenarios: instant answer with causal masking; instant answer without causal masking; and $n$ pause tokens (with causal masking). Training details are in Appendix F.

**Hints.** We found that in all regimes, gradient-based training for a shallow Transformer struggles to learn parity using only the loss on the final prediction (never performing above chance at any length). Therefore, we follow Pfau et al. [30] in providing additional "hint" labels during training.

For the non-causal and pause token Transformers, this is in the form of the $n$ threshold values $t_j = \mathbb{I}[\sum_{i=1}^{n} x_i \geq j]$ for all $j \in [1, n]$. This hint is analogous to operations performed by the first layer of the reference 2-layer threshold circuit for parity. When training with pause tokens, we compute the loss of the final prediction, and with respect to the hint over the pause tokens, as depicted below. This additional supervision directly encourages parallel threshold-like computations

| Input | $x_1$ | ... | $x_n$ | -1 | ... | -1 | END |
|---|---|---|---|---|---|---|---|
| Labels | $\square$ | ... | $\square$ | $t_1$ | ... | $t_n$ | $\texttt{PARITY}_n$ |
| Loss mask | 0 | ... | 0 | 1 | ... | 1 | 1 |

When training the non-causal Transformer without pause tokens, we use the same hints, but located as below, with loss computed over all tokens.

| Input | $x_1$ | ... | $x_n$ | END |
|---|---|---|---|---|
| Labels | $t_1$ | ... | $t_n$ | $\texttt{PARITY}_n$. |

For the causal Transformer, the hint has to be serial, as it cannot compute threshold values for the entire sequence at any token but the last. We provide serial hints in the form of subparities: for token $x_j$, the label is the parity until index $j$, $p_j = \texttt{PARITY}_j$. In the instant answer regime, the loss is computed over the entire sequence as below.

| Input | $x_1$ | ... | $x_n$ | END |
|---|---|---|---|---|
| Labels | $p_1$ | ... | $p_n$ | $\texttt{PARITY}_n$. |

**Results.** We can see in Figure 2 that with causal masking and no pause tokens, the accuracy of the Transformer on parity falls to random guessing by the time the sequence is 100 bits long. The removal of the causal mask allows the Transformer to achieve near-perfect accuracy until the sequence reaches 200 bits, providing support for our hypothesis that causal masking leads to a bottleneck when the Transformer needs global information. However, pause tokens provide a non-trivial gain above and beyond removing the causal mask, suggesting that there are additional benefits over providing extra tokens for global information aggregation, even for the case of parity where the required circuit size should simply be linear in $n$. Specifically, we believe these pause tokens can act as dedicated "workspace" positions that simplify how the Transformer routes and aggregates information across the

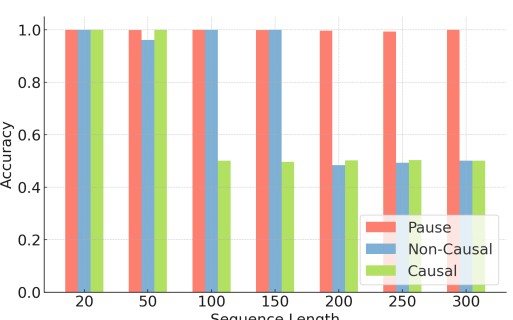

Figure 2: Test accuracy on predicting the parity of a sequence for Transformers with learned positional encodings, with and without pause tokens and causal masking. Averaged over 3 random seeds.

sequence. Having separate tokens for intermediate or partial computations can reduce the complexity of attention patterns in a way that is not possible without them.

Our empirical results suggest that, in causally masked Transformers, if a task requires global information, pause tokens can allow it to learn the task more easily, provided an appropriate signal is provided during training. This could help explain improvements on tasks like question-answering and mathematical reasoning, where the entire input is relevant to the final answer.

## 6  Discussion

**The role of pause tokens.** Our findings suggest that pause tokens can enhance the computational expressivity of Transformers, particularly in the constant precision setting. In constant precision settings, pause tokens allow Transformers to express all of $\mathsf{AC}^0$, and in logarithmic precision settings,

this increases to $\mathsf{TC}^0$. This provides a theoretical underpinning to the empirical results of Goyal et al. [14] and Pfau et al. [30].

It is known that CoT with a polynomial number of steps allows Transformers to compute everything in $\mathsf{P}$ (uniformly and $\mathsf{P/poly}$ non-uniformly). It is not proven, but it is widely believed by complexity theorists that $\mathsf{TC}^0 \subsetneq \mathsf{P/poly}$, which would imply that log-precision Transformers with CoT are strictly more expressive than those with pause tokens. In fact, a Transformer with CoT only requires depth 2 to compute anything in $\mathsf{P/poly}$ [22], which suggests that for a fixed depth, it is strictly more powerful. However, CoT requires running multiple forward passes through the Transformer (up to $\mathrm{poly}(n)$ for full $\mathsf{P/poly}$ expressivity), while pause tokens provide their additional expressivity within a single forward pass. This suggests that the two mechanisms may be complementary in improving Transformer performance, with pause tokens offering a more parallel form of computation.

That said, the practical utility of pause tokens appears limited. The theoretical gains require a polynomial number of pause tokens, and in practice models struggle to exploit this capacity effectively. These factors likely explain the weak empirical results reported by Burtsev et al. [5] and Lanham et al. [21]. Pause tokens can only help when the task demands additional computational "width" rather than depth, may require orders of magnitude more pause tokens than input tokens, and empirically need specific training strategies to become useful.

**Chain-of-thought faithfulness.** A natural concern is whether pause tokens introduce an additional layer of obfuscation in Transformer reasoning. Some prior work has suggested that chain-of-thought (CoT) reasoning can be misleading, in the sense that models may generate intermediate reasoning steps that are not causally linked to the final prediction [37]. Given that our results establish a theoretical advantage of pause tokens in expressivity, it raises the question of whether a model could use such tokens to obfuscate computations.

Our findings suggest that the risk of pause tokens being used for deceptive reasoning is limited. Unlike CoT, pause tokens are only useful for a more restricted class of problems (under the conjecture that $\mathsf{TC}^0 \subsetneq P$), and training data for reasoning tends to be in the form of sequential computation steps, rather than parallelizable as would be necessary to utilize pause tokens. It seems likely that the default learned behaviour of a Transformer, especially one trained via next-token prediction, will be to generate tokens that are meaningful to the problem it is trying to solve. Even in the case that a model does learn to make use of intermediate filler tokens, Bharadwaj [2] demonstrates that non-obfuscated tokens can be recovered from the intermediate layers.

**Quantization.** Our results offer insights into how quantization affects Transformer expressivity. Specifically, we show that if both weights and activations are quantized to constant precision, the model is limited to $\mathsf{AC}^0$ expressivity. However, if only weights are quantized while activations retain logarithmic precision, the Transformer remains in $\mathsf{TC}^0$, with the caveat that we need exact thresholding in the feedforward layers.

This follows from our proof that $\mathsf{TC}^0 \subseteq \mathsf{TF}[L, L, P]$ (Theorem C.7), which does not rely on weights having logarithmic precision except to perform thresholding in the feedforward network. The self-attention mechanism does not impose an expressivity constraint with constant precision weights (and saturation arithmetic), as long as the embedding dimension and activation precision remain log-size. This suggests that in mixed precision Transformer models, the expressivity bottleneck may be in the non-linearity of the feedforward network. If threshold-like behaviour can be implemented at sufficient precision, the model retains $\mathsf{TC}^0$ expressivity even with weight quantization.

## 7   Future directions and limitations

Our theoretical results rely on specific choices in modeling Transformer computation, particularly regarding numerical precision and arithmetic operations. Here, we outline future directions for refining these results and addressing some of the limitations.

**Saturation arithmetic and circuit complexity assumptions.** Our proof that $\mathsf{TF}[1, L, P] = \mathsf{AC}^0$ relies on the use of saturation arithmetic to allow the Transformer to attend to a polynomial number of locations when simulating $\mathsf{AC}^0$ gates with polynomial fan-in. While this assumption is a reasonable approximation to practical implementations of quantized neural networks, changing the arithmetic definition can change the expressivity of the models. It is possible to replace saturation arithmetic

with modular arithmetic and still demonstrate that $\mathsf{AC}^0 \subseteq \mathsf{TF}[1, L, P]$, but this breaks the equivalence as modular addition is not in $\mathsf{AC}^0$ (in this case the Transformer belongs to $\mathsf{ACC}^0$). Further work could analyse the expressivity under different numerical assumptions.

**Tighter bounds on the expressivity of $\mathsf{TF}_l$.** For the non-uniform setting, we are able to prove a strict separation between $\mathsf{TF}[1, L, 0]$ and $\mathsf{TF}[1, L, P]$. However, in the uniform fixed-depth setting, our separation requires a quasi-polynomial number of pause tokens. Stricter upper and lower bounds on the expressivity of fixed-depth Transformers could lead to stronger separations between uniform Transformers with and without pause tokens. In the logarithmic precision case, a tighter bound might enable the use of the depth 2 $\mathsf{TC}^0$ super-linear gate and super-quadratic wire lower bounds of Kane and Williams [20] to separate very shallow Transformers with and without pause tokens.

**Separating $\mathsf{TF}[L, L, 0]$ and $\mathsf{TF}[L, L, P]$.** While we prove that $\mathsf{TF}[L, L, P] = \mathsf{TC}^0$, we are unable to show that $\mathsf{TF}[L, L, 0] \subsetneq \mathsf{TF}[L, L, P]$. This is primarily due to the lack of known superpolynomial lower bounds for $\mathsf{TC}^0$ circuits, a long-standing open problem in circuit complexity. If such bounds were established, they could lead to a provable separation between Transformers with and without pause tokens in the logarithmic-precision regime. Alternatively, perhaps approaching the problem from the Transformer side could provide some intuition for improving $\mathsf{TC}^0$ lower bounds. Empirically, our results suggest that pause tokens can significantly improve the ability of causal Transformers to learn global functions, but we leave formal separation results as an open problem.

**Expressivity vs. learnability.** Our work focuses entirely on what Transformers with pause tokens can compute, but provides no theoretical results on whether they can learn to use this additional expressivity effectively. Empirically, we (and Pfau et al. [30]) observe that pause tokens can improve performance, but only when parallelizable explicit supervision is provided. An avenue for further research is to theoretically analyse the learnability of Transformers with pause tokens, to determine whether gradient-based training can access the full expressivity benefits.

# Acknowledgments

The authors would like to thank Hanlin Ren for illuminating discussions on several circuit complexity results. CL was supported by the Engineering and Physical Sciences Research Council, grant number EP/W524311/1.

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

# A  Further related work

**Communication complexity.** A related line of work has considered Transformers from a communication complexity perspective, though these results are mostly applicable to single-layer models. Peng et al. [29] use communication complexity techniques to show limitations in the representational power of single-layer Transformers, while Sanford et al. [33] and Bhattamishra et al. [4] use them to contrast the expressivity of Transformers and recurrent models. More recently, Chen et al. [7] have extended some of these results to the multi-layer case.

**Formal languages and automata.** Many papers have analysed Transformers through the lens of formal languages [3, 34, 26, 11], attempting to determine the class of languages that they can recognise. Liu et al. [24] proved theoretically that Transformers with $O(\log n)$ depth can simulate all finite-state automaton on inputs of length $n$, and so require depth that increases (mildly) with input length to capture all regular languages.

# B  Logspace uniformity and circuit descriptions

A logspace Turing machine may not have the space to write down its output, if the output size is $\Omega(\log n)$. For this reason, we instead require that the TM can compute any desired bit of the output in logarithmic space, i.e. it is *implictly logspace computable* [1].

**Definition B.1.** (Implicitly logspace computable) A function $f : \{0,1\}^* \to \{0,1\}^*$ is implicitly logspace computable if the mapping $x, i \mapsto f(x)_i$ can be computed by a Turing machine using logarithmic space.

**Definition B.2.** (Logspace uniformity) A circuit family $\{C_n\}$ is logspace-uniform if there is an implicitly logspace computable function mapping $1^n$ to the description of the circuit $C_n$.

**Definition B.3.** (Polylogspace uniformity) A circuit family $\{C_n\}$ is logspace-uniform if there is an implicitly polylogspace computable function mapping $1^n$ to the description of the circuit $C_n$.

We will define circuit descriptions $\mathrm{Desc}(C_n)$ for $C_n$ as a topologically ordered string of vertices, $\mathrm{Vertex}(1), \ldots, \mathrm{Vertex}(|C_n|)$, where the first $n$ vertices correspond to input variables $\mathrm{Inp}(1), \ldots, \mathrm{Inp}(n)$, and the remaining vertices represent logic gates. For $\mathsf{AC}^0$ circuits, each gate vertex $\mathrm{Vertex}(i)$ is defined by the string:

1. **Arguments:** $\mathrm{Args}(i, j)$, which specifies that the output of vertex $j$ serves as an input to vertex $i$.
2. **Type:** $\mathrm{Type}(i)$, indicating the gate's operation (AND, OR, NOT).

For $\mathsf{TC}^0$ circuits, $\mathrm{Vertex}(i)$ is defined by the string:

1. **Arguments:** $\mathrm{Args}(i, j)$, which specifies that the output of vertex $j$ serves as an input to vertex $i$.
2. **Direction:** $\mathrm{Dir}(i)$, which species if the threshold gate is $>$ or $<$.
3. **Threshold:** $\mathrm{Thresh}(i)$, a numerical value $\theta$ representing the threshold for activation of the gate.

The topological order ensures that if we computed the circuit in order from its description, all inputs to a vertex would already have been computed by the time the vertex itself is evaluated. We will define the length of the description $|\mathrm{Desc}(C_n)|$ as the total number of input, argument, and type or threshold strings.

# C  Transformer-circuit equivalence

In this section, we will demonstrate the equivalence between Transformers with pause tokens, and the circuit classes $\mathsf{AC}^0$ and $\mathsf{TC}^0$. We will demonstrate the equivalence in the uniform case, but all the proofs still hold if these uniformity requirements are relaxed.

## C.1 Constant precision

We will first consider Transformers with constant precision, $p(n) = O(1)$. While a full precision FP32 Transformer will almost always have precision $\geq \log n$, for any realistic input length $n$, more and more LLMs are being served with aggressive quantization, with 8-bit, 4-bit, and even 1.58-bit models becoming more popular. It therefore becomes necessary to understand the computational power of models under these circumstances. The constant precision regime also allows us to show strong separations between Transformers with and without pause tokens, due to the existence of universal $\mathsf{AC}^0$ lower bounds.

**Theorem C.1.** $\mathsf{AC}^0 \subseteq \mathsf{TF}[1, L, P]$

*Proof.* Consider any logspace uniform $\mathsf{AC}^0$ circuit family $\{C_n\}$. Given a Turing machine that generates the description of $\{C_n\}$, we can modify it to instead generate the position encodings and parameters of a constant precision Transformer family $\{T_n\}$ that simulates the circuits, using $b(n) \leq |\mathrm{Desc}(C_n)|$ pause tokens.

**Positional encodings.** We will construct a positional encoding for every input symbol $\mathrm{Inp}(i)$, argument symbol $\mathrm{Arg}(i, j)$, and gate symbol $\mathrm{Type}(i)$. All encodings have dimension $d(n) = O(\log n)$, and we will use the notation $\mathbf{bin}(i)$ to refer to the binary vector representation of $0 \leq i \leq b(n)$, and $\mathbf{sbin}(i) = 2\mathbf{bin}(i) - 1$. We will use $\boldsymbol{x}^{\frown}\boldsymbol{y}$ to represent the interleaving of two vectors $\boldsymbol{x}$ and $\boldsymbol{y}$ of the same dimensionality, so that $(\boldsymbol{x}^{\frown}\boldsymbol{y})_{2i-1} = x_i$ and $(\boldsymbol{x}^{\frown}\boldsymbol{y})_{2i} = y_i$. Finally, we will define $\boldsymbol{k}(i) = \mathbf{sbin}(i)^{\frown}\mathbf{1}$, and $\boldsymbol{q}(i) = B_p(\mathbf{sbin}(i)^{\frown}(-\mathbf{1}))$. This means that

$$\boldsymbol{k}(i)^T \boldsymbol{q}(j) = \begin{cases} 0 & \text{if } i = j \\ -B_p & \text{otherwise.} \end{cases} \tag{2}$$

Post-exponentiation, the pre-normalized attention values will be 1 for $i = j$ and 0 everywhere else. We reserve 0 as a privileged index, such that for all indices $i$ in the circuit description $\boldsymbol{k}(0)^T \boldsymbol{q}(i) = -B_p$. Our position encodings will then take the following forms:

- Inputs

$$\phi(\mathrm{Inp}(i)) = (0, 0, 0, \boldsymbol{k}(i), \boldsymbol{q}(i), \boldsymbol{k}(i), \boldsymbol{q}(i)).$$

- Arguments

$$\phi(\mathrm{Arg}(i, j)) = \begin{cases} (0, 0, 1, \boldsymbol{k}(0), \boldsymbol{q}(j), \boldsymbol{k}(i), \boldsymbol{q}(i)) & \text{if } \mathrm{Type}(i) \text{ is } \mathtt{OR} \\ (1, 0, 1, \boldsymbol{k}(0), \boldsymbol{q}(j), \boldsymbol{k}(i), \boldsymbol{q}(i)) & \text{otherwise.} \end{cases}$$

- Gates

$$\phi(\mathrm{Type}(i)) = \begin{cases} (0, 1, 0, \boldsymbol{k}(i), \boldsymbol{q}(i), \boldsymbol{k}(0), \boldsymbol{q}(i)) & \text{if } \mathrm{Type}(i) \text{ is } \mathtt{AND} \\ (0, 0, 0, \boldsymbol{k}(i), \boldsymbol{q}(i), \boldsymbol{k}(0), \boldsymbol{q}(i)) & \text{otherwise.} \end{cases}$$

We will denote the positional encoding corresponding to circuit $C_n$ as $\phi(C_n)$. Given a description of $C_n$ in prefix form, $\phi(C_n)$ has the following structure

$$\phi(C_n) = \left[ \phi(\mathrm{Inp}(1)) \ldots \phi(\mathrm{Inp}(n))\phi(\mathrm{Arg}(n+1, j)) \ldots \phi(\mathrm{Arg}(n+1, k))\phi(\mathrm{Type}(n+1)) \ldots \phi(\mathrm{Type}(|C_n|)) \right]$$

The input to the Transformer is then a binary string $x$ of length $n$, with $b(n) = \mathrm{poly}(n)$ pause tokens appended to it. We choose to use 0 for the pause tokens to simplify the construction, but any fixed $x \in \mathbb{F}_p$ will work, as it can be corrected for in the feedforward layers. We append the position encodings at each position, to give vectors in $O(\log n)$. In matrix form, the encoded input $X$ appears as

$$X = \begin{bmatrix} x & | & 0 \\ \hline & \phi(\mathrm{Desc}(C_n)) & \end{bmatrix}$$

**Transformer layers.** For each individual layer in the $\mathsf{AC}^0$ circuit, we will construct 2 (causally masked) Transformer layers that perform the same computation. For an $\mathsf{AC}^0$ circuit of depth $l$, this will result in a Transformer with $2l$ (and hence constant) layers.

The proof proceeds by induction. If we assume that the first $l$ layers of the circuit have been computed correctly, then at each $\mathrm{Type}(i)$ token location in the input preceding layer $l+1$, we have the computed value of $\mathrm{Vertex}(i) = v_i$ (this is $x_i$ for $\mathrm{Inp}(i)$, and the recursively computed value for $\mathrm{Type}(i)$). Each argument token $\mathrm{Arg}(i,j)$ will have value $0$. For $\mathrm{Type}(i)$ tokens in layers $> l$ the value at this point can be arbitrary (denoted by $\square$), as it will be overwritten. For convenience, we will denote the set of vertices in the first $l$ layers of the circuit $C_n$ by $C_n[\leq l]$. We will demonstrate that the next two Transformer layers correctly compute the output values for the gates in layer $l+1$.

The first attention layer copies previously computed vertex values into the correct argument positions for layer $l+1$. We construct $W^{QK}$ such that query

- $\mathrm{Inp}(i)$ attends only to itself.
- $\mathrm{Arg}(i,j)$ attends only to $\mathrm{Vertex}(j)$.
- $\mathrm{Type}(i)$ attends only to itself.

After softmax and normalization, we will have an attention value of $1$ on the token $\boldsymbol{u}$ it attends to, and $0$ everywhere else. The value matrix $W^V$ will pass through the first index of the token attended to ($\boldsymbol{u}_1$), and zero out everything else (explicit matrices for $W^{QK}$ and $W^V$ can be found in Appendix C.3.1). Concretely, after the attention layer and residual connection, for each token type we will have

$$\mathrm{Inp}(i) \leftarrow 2v_i \tag{3}$$

$$\mathrm{Arg}(i,j) \leftarrow \begin{cases} v_j & \text{if } \mathrm{Vertex}(j) \in C_n[\leq l] \\ \square & \text{otherwise.} \end{cases} \tag{4}$$

$$\mathrm{Type}(i) \leftarrow \begin{cases} 2v_i & \text{if } \mathrm{Vertex}(i) \in C_n[\leq l] \\ \square & \text{otherwise.} \end{cases} \tag{5}$$

We get the factors of $2$ due to self-attention and residual connections, but these are corrected for in the feedforward network. We apply a constant-depth feedforward network to each token $\boldsymbol{u}$, computing $\boldsymbol{u}_1 \leftarrow \mathbb{I}[(\boldsymbol{u}_1 - \boldsymbol{u}_2) \neq 0] - \boldsymbol{u}_1$, $\boldsymbol{u}_{i\neq 1} \leftarrow 0$ (for the ReLU network construction, see Appendix C.3.2). $\boldsymbol{u}_2$ is $1$ if $\boldsymbol{u}$ corresponds to $\mathrm{Arg}(i,j)$ and $\mathrm{Type}(i)$ is AND or NOT, and $0$ otherwise. Therefore, post feedforward and residual connection, we have

$$\mathrm{Inp}(i) \leftarrow v_i \tag{6}$$

$$\mathrm{Arg}(i,j) \leftarrow \begin{cases} v_j & \text{if } \mathrm{Type}(i) \text{ is OR and } \mathrm{Vertex(j)} \in C_n[\leq l], \\ 1 - v_j & \text{if } \mathrm{Type}(i) \text{ is AND or NOT and } \mathrm{Vertex(j)} \in C_n[\leq l], \\ \square & \text{otherwise.} \end{cases} \tag{7}$$

$$\mathrm{Type}(i) \leftarrow \begin{cases} v_i & \text{if } \mathrm{Vertex}(i) \in C_n[\leq l] \\ \square & \text{otherwise.} \end{cases} \tag{8}$$

The positional encodings of all tokens are unaffected by these layers, as they zero out the positional information, and the residual connections copy it over.

The second Transformer layer then computes the gate outputs for layer $l + 1$. Due to our use of saturation arithmetic, we can attend an arbitrary subset of input locations with constant precision. The maximum value of the normalizing constant in the attention computation is $B_p$, and post-exponentiation attention values are either 0 or 1, so our minimal non-zero attention value is $1/B_p \geq 1/2^p$, and so is still representable as a non-zero value. $W^{QK}$ is constructed such that query

- $\text{Inp}(i)$ attends only to itself.

- $\text{Arg}(i, j)$ attends only to itself.

- $\text{Type}(i)$ attends to all $\text{Arg}(i, \cdot)$, and does not attend to itself.

$W^{PV}$ again only allows through $\boldsymbol{u}_1$. If we define $m_i$ as the input arity of $\text{Vertex}(i)$, and define $\hat{m}_i = [m_i]_p$, then the representation of each token after the attention layer and residual connection becomes

$$\text{Inp}(i) \leftarrow 2v_i \tag{9}$$

$$\text{Arg}(i, j) \leftarrow \begin{cases} 2v_j & \text{if Type}(i) \text{ is OR and Vertex(j)} \in C_n[\leq l], \\ 2 - 2v_j & \text{if Type}(i) \text{ is AND or NOT and Vertex(j)} \in C_n[\leq l], \\ \square & \text{otherwise.} \end{cases} \tag{10}$$

$$\text{Type}(i) \leftarrow \begin{cases} \frac{1}{\hat{m}_i} \sum_{\{j \,|\, \text{Arg}(i,j)\}} v_j & \text{if Type}(i) \text{ is OR and Vertex(i)} \in C_n[\leq l + 1], \\ \frac{1}{\hat{m}_i} \sum_{\{j \,|\, \text{Arg}(i,j)\}} (1 - v_j) & \text{if Type}(i) \text{ is AND or NOT and Vertex(i)} \in C_n[\leq l + 1], \\ \square & \text{otherwise.} \end{cases} \tag{11}$$

The addition is iterated fixed point addition, but all we require is that the result will be non-zero if any of the addition terms is non-zero. The value of $\text{Type}(i)$ will be $> 0$ if it is: OR and any argument $v_j = 1$; AND and any argument $v_j = 0$; NOT and the argument $v_j = 0$. Otherwise, the value will be 0. The feedforward network will then compute

$$\boldsymbol{u}_1 \leftarrow \begin{cases} -\boldsymbol{u}_1 & \text{if } \boldsymbol{u}_4 = 1 \ (\boldsymbol{u} \text{ is an Arg token}), \\ 1 - \mathbb{I}[\boldsymbol{u}_1 > 0] - \boldsymbol{u}_1 & \text{if } \boldsymbol{u}_4 = 0 \text{ and } \boldsymbol{u}_3 = 1 \ (\boldsymbol{u} \text{ is Type}(i) \text{ with type AND}), \\ \mathbb{I}[\boldsymbol{u}_1 > 0] - \boldsymbol{u}_1 & \text{otherwise.} \end{cases} \tag{12}$$

$$\boldsymbol{u}_{i \neq 1} \leftarrow 0 \tag{13}$$

where $u$ is any token. Combined with the residual connection, this feedforward layer sets $\boldsymbol{u}_1$ to 1 if $\boldsymbol{u}_1 = 0$ and $\boldsymbol{u}$ corresponds to an AND gate, 0 if $\boldsymbol{u}$ is an Arg token, and $\mathbb{I}[\boldsymbol{u}_1 > 0]$ for all other tokens. We can see that for each token $\text{Type}(i)$ in circuit layers $\leq l + 1$, if it is

- NOT: the first Transformer layer copies the arguments to the correct position and computes NOT $(1 - v_j)$, and the second layer copies it to the gate location.

- OR: the first layer copies all arguments to the correct position, and the second layer computes OR by summing the inputs and returning 1 only if the sum is $> 0$.

- AND: the first layer copies all arguments to the correct position and computes their NOT $(1 - v_j)$, and the second layer computes the negation of the OR of the negated arguments, by returning 1 only if the sum $= 0$. By De Morgan's laws, this is an AND gate.

Thus, following the evaluation of these two Transformer layers, the residual stream will now contain the value of all the gates in layers $\leq l + 1$ of the $\text{AC}^0$ circuit, the input tokens are unchanged, and all arguments have value 0, as required to compute the next layer. Evaluating all $2l$ layers will result in the final value of $C_n$ being output at the final position.

$\square$

**Lemma C.2.** *[27] Let $f : \{0,1\}^* \to \{0,1\}^m$ be a linear space computable function. There exists a Turing machine that, for all $n \in \mathbb{N}$ and $c \in \mathbb{R}+$, uses at most $c \log n + \log m$ space to map input $1^n$ to an $\mathsf{AC}^0$ circuit of size at most $nc + c \log n + m$ and depth 3 that computes $f$ on inputs of size at most $c \log n$.*

**Lemma C.3.** *(Li et al. [22]) For any fixed $p \in \mathbb{N}$, $\mathrm{sum}_p : (\mathbb{F}_p)^n \to \mathbb{F}_p$ has uniform $\mathsf{AC}^0$ circuits.*

*Remark.* While Li et al. [22] do not explicitly state that their construction is uniform, their proof hinges on the fact that constant precision saturation arithmetic is computable by a counter-free automata. Such automata recognize star-free languages, which are definable in FO[<] [25]. Since FO[<] corresponds to DLOGTIME-uniform $\mathsf{AC}^0$ [19], and DLOGTIME $\subset L$, the construction is logspace uniform.

**Lemma C.4.** *Exponentiation $[e^x]_p$, multiplication $[xy]_p$, and division $[x/y]_p$ of constant precision numbers $x, y \in \mathbb{F}_p$ has uniform $\mathsf{AC}^0$ circuits.*

*Proof.* These functions are finite and can therefore be implemented as a constant-size lookup table. A logspace TM can trivially hardcode this constant-size circuit description. $\qquad\square$

**Theorem C.5.** $\mathsf{TF}[1, L, P] \subseteq \mathsf{AC}^0$

*Proof.* We demonstrate that every logspace uniform Transformer with logarithmic-dimensional embeddings, constant precision, and polynomial pause tokens can be simulated by a uniform circuit family. The proof proceeds by decomposing the Transformer's operations into components computable in $\mathsf{AC}^0$.

**Positional encodings.** By construction, our positional encodings can be computed in logarithmic space, with binary inputs $i$ and $n$ of dimension at most $k \log n$, and outputs of dimension $m = k' \log n$. We can use logarithmic space to construct a counter maintaining the index $i$ (from 1 to $b(n)$) and construct constant-size circuits to convert $i$ into binary. By Lemma C.2 we can construct a logspace Turing machine mapping $1^n$ to the description of an $\mathsf{AC}^0$ circuit computing $\phi(i, n)$.

**Attention layers.** We will prove that the action of a single head can be computed in uniform $\mathsf{AC}^0$, and as our model uses a constant number of heads, the action of the entire layer is in uniform $\mathsf{AC}^0$. Recall that the attention computation is defined as

$$\boldsymbol{x}_i \leftarrow W^V \sum_{j=1}^{i} \frac{\exp(f(\boldsymbol{x}_i, \boldsymbol{x}_j))}{Z_i} \boldsymbol{x}_j \tag{14}$$

$$f(i, j) = \boldsymbol{x}_i^T W^{QK} \boldsymbol{x}_j \tag{15}$$

$$Z_i = \sum_{j=1}^{i} \exp(f(\boldsymbol{x}_i, \boldsymbol{x}_j)) \tag{16}$$

First, we demonstrate that the attention function $f$ has uniform $\mathsf{AC}^0$ circuits. By definition $W^{QK}$ is logspace computable, and so we can determine any entry $W_{ij}^{KQ}$ by simulating $M_1$ using logspace. $f$ can therefore be computed in logspace, by using a counters $k$ and $l$ over the length of the vectors, computing each $W_{kl}^{KQ}(\boldsymbol{x}_j)_k(\boldsymbol{x}_i)_l$, and storing the summation of these terms in constant space. By Lemma C.2, $f$ is in uniform $\mathsf{AC}^0$. To do this for all query and key vectors, we can use two counters over the token locations.

Exponentiation of these attention values and multiplication with $x_j$ can be done in uniform $\mathsf{AC}^0$ by Lemma C.4. Summation over the (up to $\mathrm{poly}(n)$) attention values for calculating the normalizing constant and summation of the value vectors can be done in uniform $\mathsf{AC}^0$ by Lemma C.3.

As $W^V$ is logspace computable, we can construct a logspace computable function $g$ to compute $W^V x$, again by using counters over the indices of $W^V$. This function has a logarithmic output dimension, and so we can use Lemma C.2 to prove that $g$ has a uniform $\mathsf{AC}^0$ circuit.

**Feedforward layers.** Each feedforward layer involves multiplying a token vector with a logspace computable weight matrix. Similarly to $g$, this is in uniform $\mathsf{AC}^0$.

As we have demonstrated that all Transformer computations for each layer are in uniform $\mathsf{AC}^0$, and we have a constant number of layers, $\mathsf{TF}[1, L, P] \subseteq \mathsf{AC}^0$.

$\square$

**Corollary C.6.** $\mathsf{TF}[1, L, P] = \mathsf{AC}^0$

## C.2 Logarithmic precision

**Theorem C.7.** $\mathsf{TC}^0 \subseteq \mathsf{TF}[L, L, P]$.

*Proof.* Consider any logspace uniform $\mathsf{TC}^0$ circuit family $\{C_n\}$. Given a Turing machine that generates the description of $\{C_n\}$, we can modify it to instead generate the position encodings and parameters of a constant precision Transformer family $\{T_n\}$ that simulates the circuits, using $b(n) \le |\mathrm{Desc}(C_n)|$ pause tokens.

**Positional encodings.** Define $\boldsymbol{k}(i)$ and $\boldsymbol{q}(i)$ as in the $\mathsf{AC}^0$ case, though $B_p$ now refers to the largest representable number with logarithmic precision. The positional encodings are:

- Inputs

$$\phi(\mathrm{Inp}(i)) = (0, 0, 0, \boldsymbol{k}(i), \boldsymbol{q}(i), \boldsymbol{k}(i), \boldsymbol{q}(i)).$$

- Arguments

$$\phi(\mathrm{Arg}(i, j)) = \begin{cases} (0, 0, 1, \boldsymbol{k}(0), \boldsymbol{q}(j), \boldsymbol{k}(i), \boldsymbol{q}(i)) & \text{if } \mathrm{Dir}(i) \text{ is } > \\ (1, 0, 1, \boldsymbol{k}(0), \boldsymbol{q}(j), \boldsymbol{k}(i), \boldsymbol{q}(i)) & \text{otherwise.} \end{cases}$$

- Gates

$$\phi(\mathrm{Thresh}(i)) = \begin{cases} (0, \theta/m_i, 0, \boldsymbol{k}(i), \boldsymbol{q}(i), \boldsymbol{k}(0), \boldsymbol{q}(i)) & \text{if } \mathrm{Dir}(i) \text{ is } > \\ (0, -\theta/m_i, 0, \boldsymbol{k}(i), \boldsymbol{q}(i), \boldsymbol{k}(0), \boldsymbol{q}(i)) & \text{otherwise,} \end{cases}$$

where $\theta$ is the threshold value and $m_i$ is the input arity of the vertex.

$\phi(C_n)$ has the following structure

$$\phi(C_n) = \left[ \begin{array}{ccccc} | & | & | & | & | & | \\ \phi(\mathrm{Inp}(1))\dots\phi(\mathrm{Inp}(n)) & \phi(\mathrm{Arg}(n+1, j))\dots\phi(\mathrm{Arg}(n+1, k) & \phi(\mathrm{Thresh}(n+1))\dots\phi(\mathrm{Threshold}(|C_n|)) \\ | & | & | & | & | & | \end{array} \right]$$

**Transformer layers.** Again, we use two Transformer layers per circuit layer, and proceed by induction, assuming that all $l$ previous circuit layers have been correctly computed.

The first attention layer performs the same operation as in the $\mathsf{AC}^0$ case, copying previously computed vertices to argument locations. The first feedforward layer computes

$$\boldsymbol{u}_1 \leftarrow \begin{cases} \mathbb{I}[\boldsymbol{u}_1 > 0] - \boldsymbol{u}_1 & \text{if } \boldsymbol{u}_2 = 0 \\ -\mathbb{I}[\boldsymbol{u}_1 > 0] - \boldsymbol{u}_1 & \text{otherwise.} \end{cases} \tag{17}$$

$$\boldsymbol{u}_{i \ne 1} \leftarrow 0. \tag{18}$$

Post feedforward and residual connection, we have

$$\text{Inp}(i) \leftarrow v_i \tag{19}$$

$$\text{Arg}(i,j) \leftarrow \begin{cases} v_j & \text{if } \text{Dir}(i) \text{ is} > \text{ and Vertex}(j) \in C_n[\leq l], \\ -v_j & \text{if } \text{Dir}(i) \text{ is} < \text{ and Vertex}(j) \in C_n[\leq l], \\ \square & \text{otherwise.} \end{cases} \tag{20}$$

$$\text{Thresh}(i) \leftarrow \begin{cases} v_i & \text{if Vertex}(i) \in C_n[\leq l] \\ \square & \text{otherwise.} \end{cases} \tag{21}$$

The second attention layer also acts in the same way as in the $\mathsf{AC}^0$ case. However, due to the logarithmic precision we no longer have potential overflow when attending to $\text{poly}(n)$ argument locations, and the attention layer will be able to attend to all arguments with attention weights summing to 1, so that averaging of the inputs is performed correctly. After the attention layer and residual connection we have

$$\text{Inp}(i) \leftarrow 2v_i \tag{22}$$

$$\text{Arg}(i,j) \leftarrow \begin{cases} 2v_j & \text{if } \text{Dir}(i) \text{ is} > \text{ and Vertex}(j) \in C_n[\leq l], \\ -2v_j & \text{if } \text{Dir}(i) \text{ is} < \text{ and Vertex}(j) \in C_n[\leq l], \\ \square & \text{otherwise.} \end{cases} \tag{23}$$

$$\text{Thresh}(i) \leftarrow \begin{cases} \frac{1}{m_i}\sum_{\{j \mid \text{Arg}(i,j)\}} v_j & \text{if } \text{Dir}(i) \text{ is} > \text{ and Vertex}(i) \in C_n[\leq l+1], \\ \frac{1}{m_i}\sum_{\{j \mid \text{Arg}(i,j)\}} -v_j & \text{if } \text{Dir}(i) \text{ is} < \text{ and Vertex}(i) \in C_n[\leq l+1], \\ \square & \text{otherwise.} \end{cases} \tag{24}$$

The second feedforward layer will compute

$$\boldsymbol{u}_1 \leftarrow \begin{cases} -\boldsymbol{u}_1 & \text{if } \boldsymbol{u}_4 = 1 \ (\boldsymbol{u} \text{ is an Arg token}) \\ \mathbb{I}[(\boldsymbol{u}_1 - \boldsymbol{u}_3) > 0] - \boldsymbol{u}_1 & \text{otherwise,} \end{cases} \tag{25}$$

$$\boldsymbol{u}_{i \neq 1} \leftarrow 0 \tag{26}$$

For every token type, we have

$$\text{Inp}(i) \leftarrow v_i \tag{27}$$

$$\text{Arg}(i,j) \leftarrow 0 \tag{28}$$

$$\text{Thresh}(i) \leftarrow \begin{cases} \mathbb{I}\left[\sum_{\{j \mid \text{Arg}(i,j)\}} v_j > k\right] & \text{if } \text{Dir}(i) \text{ is} > \text{ and Vertex}(i) \in C_n[\leq l+1], \\ \mathbb{I}\left[\sum_{\{j \mid \text{Arg}(i,j)\}} v_j < k\right] & \text{if } \text{Dir}(i) \text{ is} < \text{ and Vertex}(i) \in C_n[\leq l+1], \\ \square & \text{otherwise.} \end{cases} \tag{29}$$

These are the correct output values for the threshold gates in layers $\leq l+1$, and we have ensured that we have the required residual values to compute the next layer. Evaluating all $2l$ layers will result in the final value of $C_n$ being output at the final position.

$\square$

**Lemma C.8.** *(Merrill and Sabharwal [27], paraphrased) For any $p(n) = O(\log n)$, the function* $\text{sum}_{p(n)} : (\mathbb{F}_{p(n)})^n \to \mathbb{F}_{p(n)}$ *has uniform* $\mathsf{TC}^0$ *circuits.*

**Lemma C.9.** *[18] Multiplication and division of two $n$-bit numbers is in uniform* $\mathsf{TC}^0$.

**Lemma C.10.** *Exponentiation $[e^x]_{p(n)}$ of an $p(n) = O(\log n)$ bit number is in uniform* $\mathsf{TC}^0$.

*Proof.* By Hesse [18], iterated multiplication of $n$ $n$-bit numbers is in uniform $\mathsf{TC}^0$. As exponentiation of an $p(n) = O(\log n)$ bit number can be seen as iterated multiplication of $n$ constant bit numbers, it is in uniform $\mathsf{TC}^0$. $\square$

**Theorem C.11.** $\mathsf{TF}[L, L, P] \subseteq \mathsf{TC}^0$

The only difference between this proof and the constant precision case is that we must be able to perform the required arithmetic with logarithmic precision. By Lemma C.8, Lemma C.9 and Lemma C.10, all the necessary arithmetic operations are in uniform $\mathsf{TC}^0$.

**Corollary C.12.** $\mathsf{TC}^0 = \mathsf{TF}[L, L, P]$

## C.3 Model parameters and feedforward networks

### C.3.1 Attention weights

- First layer:

$$
W^{QK} = \begin{bmatrix}
0 & 0 & 0 & 0 & 0 & 0 & 0 & 0 \\
0 & 0 & 0 & 0 & 0 & 0 & 0 & 0 \\
0 & 0 & 0 & 0 & 0 & 0 & 0 & 0 \\
0 & 0 & 0 & 0 & 0 & 0 & 0 & 0 \\
0 & 0 & 0 & 0 & 0 & 0 & 0 & 0 \\
0 & 0 & 0 & 0 & I & 0 & 0 & 0 \\
0 & 0 & 0 & 0 & 0 & 0 & 0 & 0 \\
0 & 0 & 0 & 0 & 0 & 0 & 0 & 0
\end{bmatrix}
\qquad
W^V = \begin{bmatrix}
1 & 0 & 0 & 0 & 0 & 0 & 0 & 0 \\
0 & 0 & 0 & 0 & 0 & 0 & 0 & 0 \\
0 & 0 & 0 & 0 & 0 & 0 & 0 & 0 \\
0 & 0 & 0 & 0 & 0 & 0 & 0 & 0 \\
0 & 0 & 0 & 0 & 0 & 0 & 0 & 0 \\
0 & 0 & 0 & 0 & 0 & 0 & 0 & 0 \\
0 & 0 & 0 & 0 & 0 & 0 & 0 & 0 \\
0 & 0 & 0 & 0 & 0 & 0 & 0 & 0
\end{bmatrix}
$$

- Second layer:

$$
W^{QK} = \begin{bmatrix}
0 & 0 & 0 & 0 & 0 & 0 & 0 & 0 \\
0 & 0 & 0 & 0 & 0 & 0 & 0 & 0 \\
0 & 0 & 0 & 0 & 0 & 0 & 0 & 0 \\
0 & 0 & 0 & 0 & 0 & 0 & 0 & 0 \\
0 & 0 & 0 & 0 & 0 & 0 & 0 & 0 \\
0 & 0 & 0 & 0 & 0 & 0 & 0 & 0 \\
0 & 0 & 0 & 0 & 0 & 0 & 0 & 0 \\
0 & 0 & 0 & 0 & 0 & 0 & I & 0
\end{bmatrix}
\qquad
W^V = \begin{bmatrix}
1 & 0 & 0 & 0 & 0 & 0 & 0 & 0 \\
0 & 0 & 0 & 0 & 0 & 0 & 0 & 0 \\
0 & 0 & 0 & 0 & 0 & 0 & 0 & 0 \\
0 & 0 & 0 & 0 & 0 & 0 & 0 & 0 \\
0 & 0 & 0 & 0 & 0 & 0 & 0 & 0 \\
0 & 0 & 0 & 0 & 0 & 0 & 0 & 0 \\
0 & 0 & 0 & 0 & 0 & 0 & 0 & 0 \\
0 & 0 & 0 & 0 & 0 & 0 & 0 & 0
\end{bmatrix}
$$

These weights encode a very simple connectivity pattern and can easily be constructed in logspace.

### C.3.2 Feedforward networks

**Constant precision.** We first define a ReLU function

$$
g(a) = B_p \left( \mathrm{ReLU}(a) + \mathrm{ReLU}\left( a - \frac{1}{2^p} \right) \right). \tag{30}
$$

$B_p$ and $2^{-p}$ are the largest and smallest representable positive values in $\mathbb{F}_p$, and as $[B_p/2^p]_p = 1$, $g(a) = \mathbb{I}[a > 0]$. The first feedforward network in our Transformer computes $\mathbb{I}[(\boldsymbol{u}_1 - \boldsymbol{u}_2) \neq 0] - \boldsymbol{u}_1$, which can be done by the following ReLU function

$$
g(\boldsymbol{u}_1 - \boldsymbol{u}_3) + g(\boldsymbol{u}_3 - \boldsymbol{u}_1) - \mathrm{ReLU}(\boldsymbol{u}_1) - \mathrm{ReLU}(-\boldsymbol{u}_1), \tag{31}
$$

implementable by a 3-layer feedforward ReLU network.

The second feedforward network computes

$$
\mathrm{ReLU}(1 - g(\boldsymbol{u}_1) + \boldsymbol{u}_3 - \boldsymbol{u}_4 - 1) + \mathrm{ReLU}(g(\boldsymbol{u}_1) - \boldsymbol{u}_3 - \boldsymbol{u}_4) - \mathrm{ReLU}(\boldsymbol{u}_1) - \mathrm{ReLU}(-\boldsymbol{u}_1), \tag{32}
$$

implementable by a 4-layer feedforward ReLU network.

**Logarithmic precision.** We use the same $g$, but now defined over $\mathbb{F}_{p(n)}$, where $p(n) = O(\log n)$. The first feedforward network computes

$$-\mathrm{ReLU}(g(\boldsymbol{u}_1) + \boldsymbol{u}_2 - 1) + (\mathrm{ReLU}(g(\boldsymbol{u}_1) - \boldsymbol{u}_2) - \mathrm{ReLU}(\boldsymbol{u}_1) - \mathrm{ReLU}(-\boldsymbol{u}_1), \qquad (33)$$

implementable by a 4-layer feedforward ReLU network.

The second feedforward network computes

$$\mathrm{ReLU}(g(\boldsymbol{u}_1 - \boldsymbol{u}_3) - \boldsymbol{u}_4) - \mathrm{ReLU}(\boldsymbol{u}_1) - \mathrm{ReLU}(-\boldsymbol{u}_1), \qquad (34)$$

implementable by a 4-layer feedforward ReLU network.

# D   Separations in expressivity

We will use the notation $\mathsf{AC}^0[h(n)]$ (and $\mathsf{TC}^0[h(n)]$) to define the class of functions computable by $\mathsf{AC}^0$ ($\mathsf{TC}^0$) circuits with $O(h(n))$ gates/vertices.

## D.1   Non-uniform constant precision Transformers

**Lemma D.1.** *[22]* $\mathsf{TF}[1, L, 0] \subsetneq \mathsf{AC}^0$

*Proof (sketch).* The crux of this argument is that when the Transformer has only $n$ tokens and $d(n) = O(\log n)$ embedding dimension, it can be simulated by $\mathsf{AC}^0[n^k]$ for some fixed $k \in \mathbb{N}$, as all positional encoding, attention, and feedforward operations are computable by fixed polysize circuits. We can therefore always find some non-uniform $\mathsf{AC}^0$ circuit with size $n^{k'}$, where $k \le k'$, such that the function computed by this circuit is not in $\mathsf{TF}[1, L, 0]$. $\qquad\square$

This is no longer the case when the Transformer is allowed $\mathrm{poly}(n)$ pause tokens, as an $\mathsf{AC}^0$ circuit simulating this Transformer must be able to add $\mathrm{poly}(n) = \cup_{k \in \mathbb{N}} n^k$ terms to compute the attention layer, and so cannot be of a fixed size $n^k$.

**Corollary D.2.** *For non-uniform Transformers and* $\mathsf{AC}^0$, $\mathsf{TF}[1, L, 0] \subsetneq \mathsf{TF}[1, L, P]$.

*Proof.* By Corollary C.6 $\mathsf{TF}[1, L, P] = \mathsf{AC}^0$, and by Lemma D.1 $\mathsf{TF}[1, L, 0] \subsetneq \mathsf{AC}^0$. $\qquad\square$

## D.2   Uniform constant precision Transformers

In the uniform case, it is more difficult to construct a function $f_k$ for all $k \in \mathbb{N}$ such that $f_k \in \mathsf{AC}^0$ and $f_k \notin \mathsf{AC}^0[n^k]$. However, we can do this for fixed-depth circuits using the tight $\mathsf{AC}^0$ lower bounds for parity [13, 17].

**Lemma D.3.** *(Limaye et al. [23], paraphrased) There exists a fixed depth size hierarchy for uniform* $\mathsf{AC}^0$, *such that* $\mathsf{AC}_l^0[n^{k\epsilon}] \subsetneq \mathsf{AC}_l^0[n^k]$ *for some fixed* $0 < \epsilon < 1$.

*Proof (sketch).* This result of Hastad [17] states that any depth-$l$ $\mathsf{AC}^0$ circuit for parity of $n$ variables must have size $2^{\Omega(n^{1/l-1})}$, which is tight due to the folklore depth-$l$ $\mathsf{AC}^0$ upper bound of $2^{O(n^{1/l-1})}$. These bounds give us a separation between circuits of size $s_0 = 2^{O(n^{1/l-1})}$ and $s_0^\epsilon$ for some fixed $0 < \epsilon < 1$. The same separation holds between $s^\epsilon$ and $s$ for any $s \le s_0$, by taking the parity function over some $m \le n$ variables. If we take $s(n) = O(n^k)$ ($m = \mathrm{polylog}(n)$), we obtain a separation between $\mathsf{AC}_l^0[n^{k\epsilon}]$ and $\mathsf{AC}_l^0[n^k]$.

This separation holds in uniform $\mathsf{AC}^0$, as parity over $(k \log n)^{l-1}$ bits can be computed via a simple uniform divide-and-conquer approach that computes the parity of $O(\log n)$ bit chunks in parallel at each layer. $\qquad\square$

**Theorem D.4.** $\mathsf{TF}_l[1, L, 0] \subsetneq \mathsf{TF}_l[1, L, Q]$.

*Proof.* By a similar argument to Lemma D.1 we have that $\mathsf{TF}_l[1, L, 0] \subseteq \mathsf{AC}^0_{l'}[n^{k\epsilon}]$ for some $l', k \in \mathbb{N}$ and $0 < \epsilon < 1$. By Lemma D.3 we know that there is some function $f_n$ over suitably chosen $m = \mathrm{polylog}(n)$ bits such that $f_n \notin \mathsf{AC}^0_l[n^{k\epsilon}]$, and $f_n \in \mathsf{AC}^0_l[n^k]$.

By Theorem C.1, we know that $\mathsf{TF}_l[1, L, |\mathrm{Desc}(C_n)|]$ can compute any function computed by a circuit of depth $l/2$ with $|C_n|$ unbounded fan-in AND, OR, and NOT gates. By the parity bounds, we know that a depth $l/2$ circuit of size $|C_n| = 2^{O\left(m^{2l'/(l-2)}\right)}$ can compute parity over m input bits. For $m = \mathrm{polylog}(n)$, this is quasi-polynomial in $n$. $|\mathrm{Desc}(C_n)|$ cannot be more than quasi-polynomial in $|C_n|$, and so the number of pause tokens necessary to compute the function is still quasi-polynomial.

Therefore $f_n \notin \mathsf{TF}_l[1, L, 0]$ and $f_n \in \mathsf{TF}_l[1, LQ]$. It is intuitively obvious that $\mathsf{TF}_l[1, L, 0] \subseteq \mathsf{TF}_l[1, L, Q]$, and so the proof is complete. $\qquad\square$

### D.3 Width cannot compensate for depth

It should be clear that the addition of pause tokens does not enable the Transformer class to solve problems that require more serial computation steps, instead increasing the parallel width of computation. For finite-depth, constant precision Transformers with up to $\mathrm{poly}(n)$ pause tokens, we will always be limited to the class $\mathsf{AC}^0$. A prudent question is whether, within this class, increasing the depth of the Transformer increases its expressivity? This seems intuitively obvious, but we provide a formal proof below.

**Theorem D.5.** *For any depth $l$ there exists a constant $m > l$ such that $\mathsf{TF}_l[1, L, P] \subsetneq \mathsf{TF}_m[1, L, P]$.*

*Proof.* By Theorem C.1 and Theorem C.5, we have that $\mathsf{AC}^0_{l/2} \subseteq \mathsf{TF}_l[1, L, P] \subseteq \mathsf{AC}^0_{l'}$. By the result of Hastad [17], there exist functions in $\mathsf{AC}^0_{d+1}$ that are not in $\mathsf{AC}^0_d$ (with polynomial size). Therefore, $\mathsf{TF}_m[1, L, P]$, where $m \geq 2l'$, is strictly more expressive than $\mathsf{TF}_l[1, L, P]$. $\qquad\square$

## E  2-layer threshold circuit for parity

Let $x_1, \ldots, x_n \in \{0, 1\}$ be the inputs. Define $s = \sum_{i=1}^{n} x_i$.

**First Layer:**  For $k = 1, 2, \ldots, n$, construct a threshold gate

$$G_k = \mathbb{I}\left[\sum_{i=1}^{n} x_i \geq k\right].$$

In other words,

$$G_k = \begin{cases} 1 & \text{if } s \geq k, \\ 0 & \text{otherwise.} \end{cases}$$

Note that exactly $G_1, G_2, \ldots, G_s$ will output 1, and the gates $G_{s+1}, \ldots, G_n$ will output 0.

**Second Layer:**  Introduce one additional threshold gate $F$ that takes inputs from $G_1, \ldots, G_n$:

$$F = \mathbb{I}\left[\sum_{i=1}^{n} (-1)^{i+1} G_i \geq 1\right].$$

**Correctness:**

$$\sum_{i=1}^{n} (-1)^{i+1} G_i = \sum_{i=1}^{s} (-1)^{i+1}$$

This is a simple alternating sum of 1 and $-1$ for $s$ terms, which is 1 if $s$ is odd, and 0 if it is even. Therefore,

$$F = 1 \quad \Longleftrightarrow \quad \sum_{i=1}^{n} (-1)^{i+1} G_i \geq 1 \quad \Longleftrightarrow \quad s \text{ is odd.}$$

Thus $F$ outputs 1 exactly when the number of 1's among $x_1, \ldots, x_n$ is odd, i.e. $F$ computes the parity function.

# F   Experiment details

All experiments were conducted using a single NVIDIA V100 GPU. The model is a custom variant of Huggingface's GPT-2 implementation, to allow for different input dimensions and position encodings. Our Transformer model uses 2 layers, 4 attention heads, and a hidden dimension of 32. Positional encodings are learned during training. All models are trained for 50 epochs using the Adam optimiser with a learning rate of $5 \times 10^{-4}$, $\beta_1 = 0.9$, $\beta_2 = 0.999$, and no weight decay. We disable mixed precision and gradient clipping, as the models are small and training is stable.

Validation accuracy is monitored each epoch, and the model achieving the highest validation accuracy is used for test set evaluation. Training, validation, and test datasets consist of 500,000, 5,000, and 50,000 examples, respectively. The datasets consist of uniformly sampled bitstrings, where the label is the parity. We generate a dataset per random seed for each length in $\{20, 50, 100, 150, 200, 250, 300\}$.

All experiments are logged using Weights & Biases, including batch and epoch metrics. Models are saved every 5 epochs and upon reaching a new maximum validation accuracy. The final test accuracy reported in the plots is averaged over three random seeds.

# G   Attention pattern analysis

Here we analyse the positional attention patterns of trained models in each class of causal, non-causal, and pause tokens, to try and understand what mechanisms/algorithms they may be using to compute the parity.

To compare attention distributions across positions, we transform raw attention weights $p(k \mid q)$ for query position $q$ and key position $k$ into a *symmetric normalised form* that measures deviation from a uniform baseline while remaining bounded and zero-centred. Let $K_{\text{eff}}(q)$ denote the number of unmasked keys visible to query $q$, and define the uniform reference distribution $U(k \mid q) = 1/K_{\text{eff}}(q)$. We first form the likelihood ratio

$$r(k \mid q) = \frac{p(k \mid q)}{U(k \mid q)} = K_{\text{eff}}(q) \, p(k \mid q),$$

which quantifies how strongly query $q$ attends to key $k$ relative to a uniform baseline. To obtain a symmetric, bounded representation, we define

$$\text{SLR}(k \mid q) \;=\; \frac{K_{\text{eff}}(q) \, p(k \mid q) - 1}{K_{\text{eff}}(q) \, p(k \mid q) + 1} \;\in\; (-1, 1).$$

Here $\text{SLR}(k \mid q) = 0$ corresponds to uniform attention, positive values indicate higher-than-uniform weighting, and negative values indicate suppression relative to the baseline. This *symmetric log-ratio* (SLR) mapping preserves the ordering of the original likelihood ratios, compresses large deviations, and provides a bounded scale suitable for visual comparison across layers and heads.

In Figure 3 we can see that the unmasked and pause tokens models attend very uniformly to all tokens in the first layer, as would be expected if it were implementing a method similar to the 2-layer threshold circuit described in Appendix E. In the second layer, the first two heads in the pause token model attend largely to pause token positions (rather than input tokens), again corresponding to the circuit. However, it does not attend heavily to all the pause tokens, suggesting some divergence from this mechanism. The non-causal model attends over all tokens when combining head 1 and head 2 in layer 1, as would be expected if it were using the threshold circuit mechanism. The causal model has less structured attention patterns that elude easy analysis.

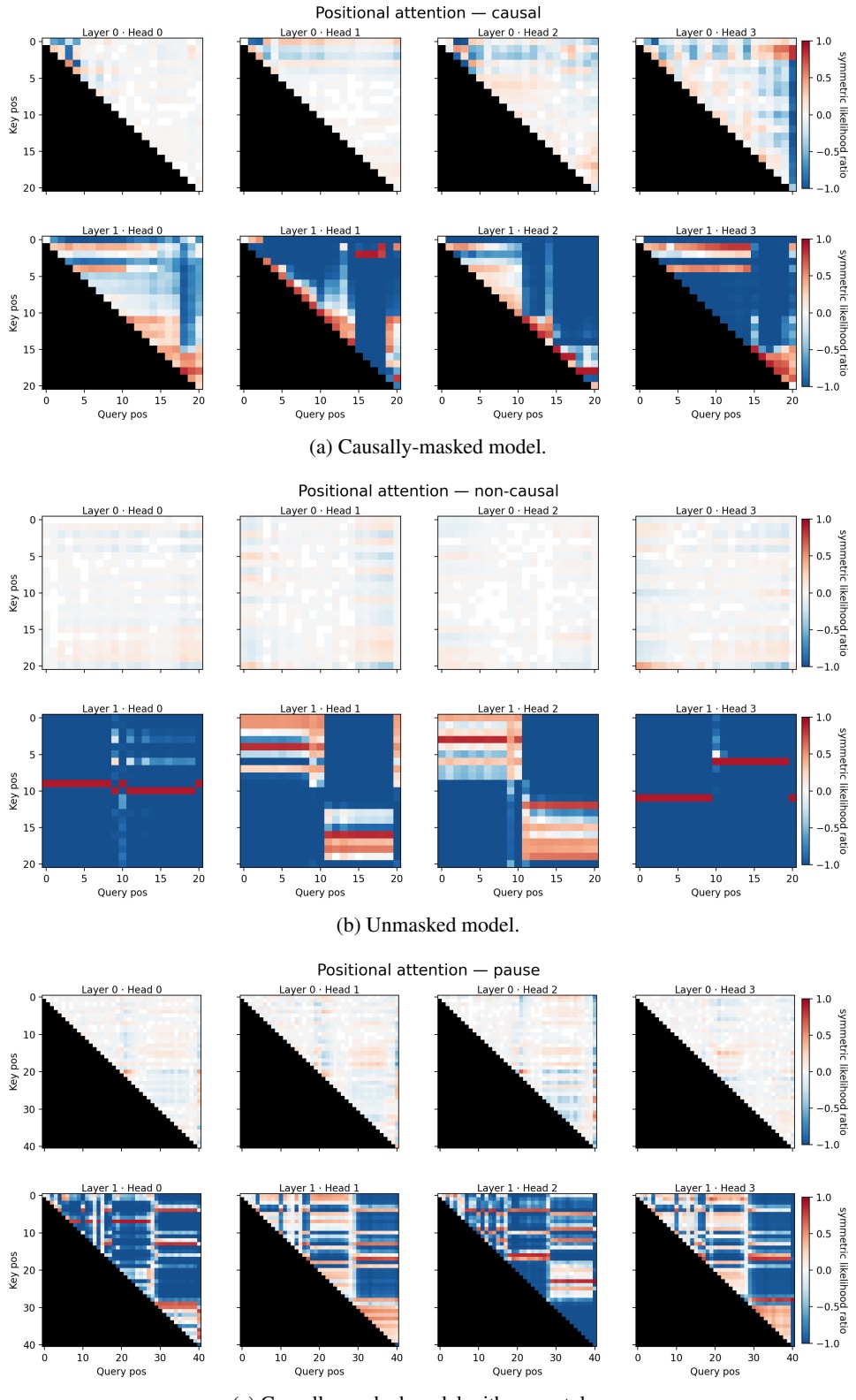

(a) Causally-masked model.

(b) Unmasked model.

(c) Causally masked model with pause tokens.

Figure 3: Symmetric attention maps, demonstrating deviation from the uniform distribution for each query position.

