# OpenReview forum: "Pause Tokens Strictly Increase the Expressivity of Constant-Depth Transformers"
_NeurIPS.cc/2025/Conference — NeurIPS 2025 poster_

### Official Review · Reviewer_MSH9 · 2025-06-26

**Clarity:** 3
**Significance:** 3
**Originality:** 3
**Rating:** 5
**Confidence:** 3

**Summary:**

This paper formally characterizes how pause tokens increase the expressivity of transformers. In the fixed precision case, the paper shows that transformers with pause tokens are equivalent to the circuit class $\mathsf{AC}^0$, and strictly more expressive than transformers without pause tokens. In the log precision case, transformers with pause tokens are equivalent to $\mathsf{TC}^0$, but the paper does not provide a separation result for transformers without pause tokens. Empirically, the paper shows that (log precision) transformers with pause tokens can solve the parity task on long inputs, where tranformers without pause tokens fail.

**Questions:**

- Would it be possible to give an example of a problem in $\text{AC}^0$ that can be solved with padding tokens but not without padding tokens?
- In the experiment section, is the non-causal model (without pause tokens) also trained with hints? Would it be possible to provide an analogous figure to the figure at the bottom of page 7?


Minor comments:

- What is $\mathcal{A}$ (in line 150)?
- It could be helpful to briefly define "advice" (in line 261), for readers who aren't familiar with the reference.

**Ethical Concerns:**

["NO or VERY MINOR ethics concerns only"]

**Final Justification:**

This paper provides new theoretical insight into an empirical phenomenon, the paper is clearly written, and the theoretical results are supported by some empirical analysis. The limitations of the paper are reasonable and clearly acknowledged by the authors. After the reading the rebuttal and the other reviews, I still recommend to accept the paper.

**Limitations:**

Yes

**Quality:**

3

**Strengths And Weaknesses:**

*Strengths*

- This paper provides theoretical insight into the empirical finding that pause tokens can improve performance on some tasks. This can help explain some empirical results that have found pause tokens to be helpful in some cases but not others, and (as the authors discuss) possibly have implications for understanding chain-of-thought and the limitations of quantized models.
- I found the writing to be very clear, and I appreciated the discussion section and the detailed discussion of areas for future work.
- Experiments show that pause tokens allow transformers to solve the parity task on long inputs, where models without pause tokens fail. There is a helpful ablation showing that the benefit of pause tokens is not simply due to overcoming the causal mask.

*Weaknesses*

- The paper does not provide a theoretical separation result between $\mathsf{TF}[L,L,0]$ and $\mathsf{TF}[L,L,P]$ (i.e., showing that log-precision transformers with pause tokens are strictly more expressive than without pause tokens). This limitation is discussed in detail and seems reasonable to me.
- The experiment (section 5) is not directly related to the theoretical results. To my understanding, a log precision transformer without pause tokens is expressive enough to compute parity (using the two layer circuit construction). The experiment shows that these models struggle to learn it in practice, while pause tokens make learning easier in practice. This result is interesting, but not directly related to the expressivity results in the paper. The authors discuss these limitations.
- It would have been interesting to see experiments with fixed precision transformers, but I understand that it is impractical to train models with precision $< O(\log n)$, which the authors discuss.

Overall, I think this paper makes a clear and valuable contribution to our theoretical understanding of transformers. I think all of the weaknesses are reasonable, and acknowledged in the paper.

---

> ### Author Rebuttal · Authors · 2025-07-31
>
> We thank the reviewer for their thoughtful and constructive comments, and are glad they found the contribution clear and valuable.
>
> **Weaknesses:**
> We are glad the reviewer agrees that the limitations are reasonable and acknowledges the difficulty in replicating asymptotic constant precision in practise. We agree that parity should be computable by a two layer Transformer without pause tokens, especially without causal masking. There may exist a separation with causal masking, but as you note, achieving this separation theoretically would require a major breakthrough in $\mathsf{TC}^0$ lower bounds.
>
> **Question 1:**
> Yes, the canonical example is the Sipser function. Most easily, this can be defined as follows.
>
> Fix a depth $d$ and a fan-in $w$ such that $n = w^d$. Then repeat the following procedure until the $d$th layer produces a single output bit.
>
> First layer: Group the inputs $x$ into blocks of $w$ consecutive bits. If $d$ is odd, take the OR of each block, if $d$ is even take the AND. This produces $w^{d-1}$ output bits as input for the next layer.
>
> Second layer: Group the $w^{d-1}$ input bits to this layer into blocks of $w$ consecutive bits. Flip the gate type used in the previous layer (AND to OR and vice versa), and apply it to each block, leaving us with $w^{d-2}$ output bits.
>
> After repeating this procedure for $d$ layers we have a single output bit which is the output of the Sipser function $f_{d,w}$.
>
> Hastad’s switching lemma tells us that a depth-d circuit needs only $O(n)$ gates to compute this function, while any circuit of depth less than $d$ needs at least $n^{\Omega(\sqrt{\log n})}$ gates.
>
> If we take a Transformer of a fixed depth without pause tokens, our Lemma D.1. states that the Transformer can be simulated by some $AC^0$ circuit of depth $d$, with no more than $n^k$ gates where $k$ is constant. Therefore, if we take the Sipser function $f_{d+1,w}$, the Transformer will be unable to compute the function without pause tokens as soon as $\sqrt{\log n} > k$. We can always construct such a function no matter the depth of the Transformer, and so this is a concrete function that cannot be computed without pause tokens.
>
> **Question 2:**
> Yes, the non-causal model is provided the same hints as the causal model trained with pause tokens. We will clarify this and add a figure into the final version.
>
> **Comments:**
> Thank you for catching this. The reference to $\mathcal{A}$ (referring to the vocabulary or alphabet of the Transformer) was an oversight, and as we only consider binary functions this should be replaced with {0, 1}.
>
> We will also add a brief definition of the complexity theoretic notion of advice in the final version.

---

> > ### Comment · Reviewer_MSH9 · 2025-08-05
> >
> > Thank you for the response and for giving the example of the Sipser function. I still recommend to accept the paper.

---

### Official Review · Reviewer_3UPU · 2025-06-26

**Clarity:** 3
**Significance:** 3
**Originality:** 3
**Rating:** 4
**Confidence:** 4

**Summary:**

In this paper, the authors theoretically studied the expressivity of constant-depth log-embedding-dimension transformers with pause tokens, which are filler symbols appended to the end of the input sequences that provide more space for computation.

For constant precision transformers, the authors prove that transformer with polynomial pause tokens is computationally equivalent to AC\^0. Moreover, for the non-uniform class of transformers, transformers without pause token are strictly less expressive than its counterpart with polynomial pause tokens; for the uniform case and for a fixed length $l$, an $l$-layer transformer without pause token is strictly less expressive than its $l$-layer counterpart with quasi-polynomial pause tokens.

For logarithmic precision transformers, the authors show it is equivalent to TC\^0 when allowing polynomial number of pause tokens, although the separation for logarithmic precision transformers with and without pause tokens still remains open.

Finally, the authors conduct experiments on learning parity function, and show it is easier to learn with pauses tokens and appropriate hints during training.

**Questions:**

1.	In Figure 2, for the causal and non-causal settings, there seem to be thresholds for each setting where the model has perfecta test accuracy below the threshold and collapse to random guess when exceeding the threshold. Also, the value of the threshold is not very small (~100 or 200). Do authors have any intuition about the value of the threshold given a fixed configuration of the transformer class?

2.	There could be two possible explanations of the results of Section 5. One is the performance gap is due to expressivity with and without pause tokens. The other is they might have same expressivity but the one with pause tokens requires much fewer samples to learn. (Similar to what authors mentioned at the end of the main text as “expressivity vs learnability). Do authors have any (indirect) proof of which explanation might be correct?

**Ethical Concerns:**

["NO or VERY MINOR ethics concerns only"]

**Final Justification:**

The additional results of the attention patterns in two layers and the discussion of CoT vs pause tokens (especially the tradeoff between the inference cost and expressivity) make sense to me. Overall, I'm leaning towards acceptance and will keep my score.

**Limitations:**

See weaknesses and questions.

**Paper Formatting Concerns:**

No major formatting concerns.

**Quality:**

4

**Strengths And Weaknesses:**

**Strengths:**

1.	This is the first theoretical work studying the benefits of pause tokens. Since pause token is empirically showed useful, it is important to (theoretically) understand its benefit.
2.	The paper provides strict separation for constant precision transformers with and without pause tokens. For log-precision transformers, the paper shows it’s equivalent to $TC\^0$.
3.	The paper is well written and easy to follow.

**Weaknesses:**

1.	For logarithmic precision transformer, the paper does not show a separation between “no pause token” and “polynomial pause token”. (although it is acceptable to me since the lower bound of $TC\^0$ is still open and might not be easy to improve)

2.	It would be interesting to investigate more in section 5. Currently, the authors only compare the test accuracy among three different settings. It would be interesting to know their respective underlying mechanisms and see why those different mechanisms lead to performance difference (does not necessarily need to be theoretical analysis since analyzing training dynamics theoretically could be much more complicated and might be beyond the scope of the current paper). This could provide a more solid evidence of the benefits of using pause token for certain TC\^0 tasks.

3.	It might also be helpful to add some discussions of the comparison of pause tokens and CoTs. In particular, it could provide a clearer picture of the theoretical gain of purely increasing the computation space (or “width”) vs allowing step-by-step intermediate computation.

---

> ### Author Rebuttal · Authors · 2025-07-31
>
> We thank the reviewer for their time and constructive comments.
>
> **On log-precision separations:**
> We agree that the lack of separation in the log precision case is a limitation that we have acknowledged in the paper. We expect that proving this separation would require significant progress in complexity theory. A (even fixed-depth) size hierarchy theorem for $\mathsf{TC}^0$ would suffice, as has been conjectured in previous papers (see Merrill et al., 2023 and Pfau et al., 2024).
>
> **On mechanisms:**
> We agree that understanding how pause tokens help during training is an important and interesting direction. At present, our empirical study in Section 5 is limited to performance outcomes and ablations. At your suggestion, we examined the learning curves and attention patterns of the trained models. The learning curves do not provide direct evidence for a specific mechanism, but they do display clear signs of grokking-like behaviour: the Transformer maintains near-random accuracy for most of training, and then suddenly transitions to perfect accuracy within a few steps.
>
> The attention patterns for the pause-token model show that in the first layer, the pause tokens attend to all input tokens, while in the second layer the output token attends to all pause tokens. This suggests the Transformer may be learning a function similar to the canonical $\mathsf{TC}^0$ circuit in Appendix E, with the first layer computing intermediate threshold sums, and the final layer combining them into parity. In the non-causal case, we see a similar pattern. In the first layer, input tokens attend broadly to all others, while in the second layer, the final token attends to all inputs. In the causal case, the patterns are less structured and harder to interpret. It would be interesting to analyse how these patterns evolve during training, and we will endeavour to include such an analysis in the final version.
>
> **On CoT vs pause tokens:**
> We currently have a brief discussion on the relative strengths of CoT and pause tokens in sections 2 and 6. We will expand on this discussion with the following points. It is known that CoT with a polynomial number of steps allows Transformers to compute everything in $\mathsf{P}$ (uniformly and $\mathsf{P/poly}$ non-uniformly). It is not proven, but it is widely believed by complexity theorists that $\mathsf{TC}^0 \subsetneq \mathsf{P/poly}$, which would imply that Transformers with CoT are strictly more expressive than those with pause tokens. In fact, a Transformer with CoT only requires depth 2 to compute anything in $\mathsf{P/poly}$ (see “Chain of Thought Empowers Transformers to Solve Inherently Serial Problems”, Li et al., 2024), which certainly suggests that for a fixed depth it is strictly more powerful. However, CoT requires running multiple forward passes through the Transformer (up to $\text{poly}(n)$ for full \mathsf{P} expressivity), while pause tokens provide their additional expressivity within a single forward pass. This suggests that they may even be complementary in improving Transformer performance.
>
> **Question 1:**
> This is a good question, and our intuition is that the threshold arises from a combination of the effective capacity of the Transformer and the number of samples required to generalise at longer lengths. Even at length 100, there are $2^100$ possible inputs, and the model sees only a very small fraction of them during training. As noted above, we observe grokking-like behaviour when the model does succeed in learning, and therefore observe either random accuracy or 100% accuracy, with little in between.
>
> To disentangle whether the effect is driven more by the sample complexity or by model capacity, we are running further experiments with wider and narrower Transformer models to see how the threshold for accuracy collapse shifts. We do not yet have conclusive results, but we aim to include them in the final version.
>
> **Question 2:**
> So far we only have speculation, and are not yet comfortable putting it in the paper. However, our hypothesis is that the form of the pause token hints reduces the sample complexity of learning parity, and so this is more a function of learnability than expressivity. It is conjectured that parity is difficult to learn because it is a high-sensitivity function (see “Why are Sensitive Functions Hard for Transformers?”, Hahn et al., 2024), where a change to a single input bit changes the output bit. By contrast, the threshold sum values that use as hints are significantly less sensitive, and so should be much easier to learn when they are given as a target. Given these threshold values, the threshold computed by the second layer of the canonical $\mathsf{TC}^0$ circuit is also significantly less sensitive, and so if the model first learns to compute the threshold sums, it should then be much easier for it to learn the overall parity. We aim to analyse this further in a follow-up work.

---

> > ### Comment · Reviewer_3UPU · 2025-08-05
> >
> > Thank the authors for the detailed response. The additional results of the attention patterns in two layers and the discussion of CoT vs pause tokens (especially the tradeoff between the inference cost and expressivity) make sense to me. Overall, I'm leaning towards acceptance and will keep my score.

---

### Official Review · Reviewer_UV3u · 2025-06-29

**Clarity:** 4
**Significance:** 3
**Originality:** 3
**Rating:** 5
**Confidence:** 5

**Summary:**

The authors study the expressivity of transformer when the input is extended by a sequence of blanks ("pause tokens"). Since positional encoding is applied to these additional tokens, the resulting transformers are more expressive. To be more precise, the authors consider Boolean functions in AC0 and TC0, well known classes defined via sequences of circuits. I give a rough description of how these classes are defined. A Boolean function is treated as a sequence of functions f_1, f_2,... (where f_n takes inputs of length n). We say it is in the class if there is a sequence of circuts, satisfying some conditions, that can be generated in uniform, space-efficient way, such that n-th circuit computes the n-th function.
In particular, the authors show that transformers with constant precision and poly number of pause tokens are equivalent to AC0 and allowing logarithmic precision gives equivalence to TC0. So for instance for a sequence of functions from AC0 or TC0 there exists a sequence of transformers, satisfying the corresponding conditions, such that n-th transformer computes the n-th function and there is a uniform, space-efficient way to generate these transformers.
What's important is that the addition of pause tokens increases expressivity in the sense described above. This happens because the authors assume positional encodings that are dependent on the input length and then for a given input length, they can take a corresponding circuit and hardcode it inside these positional encodings.

**Questions:**

Q1: Can you say something about the situation where we have a single transformer for all input lengths and add pause tokens? A partial answer to this question can be found in your proofs but perhaps you have some additional insights.

**Ethical Concerns:**

["NO or VERY MINOR ethics concerns only"]

**Final Justification:**

- Expressivity of transformers is an important theoretical topic of practical relevance. Papers in this area have been presented at NeurIPS before.
- The presented results are novel and technically sound. The proofs seem correct.

**Limitations:**

yes

**Quality:**

4

**Strengths And Weaknesses:**

Strengths:
- Expressivity of transformers is an important theoretical topic of practical relevance. Papers in this area have been presented at NeurIPS before.
- The presented results are novel and technically sound. The proofs seem correct.


Weaknesses:
- This particular contributions is probably of very limited practical relevance. While there is some sort of uniformity in this setting, it is not the kind of uniformity that is relevant from the practical point of view, i.e., having a single transformer which is capable of computing the function on inputs of arbitrary length (perhaps even with length-dependent positional encodings). Indeed, the authors give some experimental results for parity, but these are not very exciting, since generally the problem with training transformers for parity is more about length generalization and not training on fixed length inputs. That being said, it is plausible that this work will have practical impact by motivating the other authors to extend this setting to a more practical scenario.

---

> ### Author Rebuttal · Authors · 2025-07-30
>
> We thank the reviewer for their time, their generous comments, and for carefully checking the technical details.
>
> **On the question of single-model uniformity:**
> We agree this is an important question. As you note, our construction achieves uniformity in the standard circuit-theoretic sense: there exists a log-space Turing machine that generates the weights and positional encodings of the $n$-th Transformer given $n$. This does not correspond to a single Transformer computing $f_n$​ for all $n$, which would require a stronger notion of uniformity.
>
> In our view, a key technical bottleneck is the ability to address all positions in the input. Implementing arbitrary circuits requires routing information to specific locations, which in turn demands positional encodings that grow with $\log n$. Without sufficient positional resolution, we do not believe it is possible to simulate general uniform circuits, even with pause tokens. While our construction assumes access to length-dependent positional encodings, we also show (Appendix C.3) that the Transformer weights themselves are block-structured andonly the size of the blocks depends on the input length (not the value of the weights). If an upper bound $n_{\text{max}}$ on input length is known, a single Transformer with fixed weights can compute all $f_n$ for $n \leq n_{\text{max}}$​, using either length-dependent or length-independent positional encodings. In the length-independent case, the number of required pause tokens grows by an additional multiplicative factor of $n$ in our construction (but remains in $\text{poly}(n)$.
>
> We agree that further work on formalising single-model uniformity and understanding its relationship to positional encoding and expressivity would be valuable, and we will clarify this point in the final version.

---

> > ### Comment · Reviewer_UV3u · 2025-08-05
> >
> > Thanks. I continue to support the paper.

---

### Official Review · Reviewer_awq1 · 2025-07-02

**Clarity:** 3
**Significance:** 2
**Originality:** 3
**Rating:** 4
**Confidence:** 3

**Summary:**

This paper provides a theoretical investigation of the empirically observed phenomenon that adding "pause tokens"  to the input of the Transformer can improve its performance. The authors analyze the expressive power of constant-depth Transformers through the lens of circuit complexity. The paper proves that constant-precision, logarithmic-width Transformers with a polynomial number of pause tokens are computationally equivalent to the circuit class AC0. With logarithmic precision, they are equivalent to TC0. The authors demonstrate that a 2-layer causally masked Transformer, which struggles to learn the parity function, can successfully learn it when provided with pause tokens and appropriate training hints.

**Questions:**

See the Weaknesses part above.

**Ethical Concerns:**

["NO or VERY MINOR ethics concerns only"]

**Final Justification:**

The authors' rebuttal has addressed my concerns, so I am raising my score to 4.

**Limitations:**

The authors discuss the limitations of this work and point out future directions.

**Quality:**

3

**Strengths And Weaknesses:**

## Strengths
- This work is the first one to study the expressive power of Transformers with "Pause Tokens".
- This paper is well-written and structured. It clearly motivates the problem and states its contributions.

## Weaknesses
**Gap Between Theory and Practice**: The theoretical results rely on adding a polynomial number of pause tokens to simulate arbitrary circuits. However, the empirical works that motivate this paper (e.g., Goyal et al., 2024) see benefits with a much smaller, often constant, number of tokens. The paper does not fully bridge the gap between its theoretical requirements and these practical observations.

**Limited Empirical Scope**: The experiments are confined to the parity problem. While this is a canonical benchmark in complexity theory, it is a synthetic task. It remains an open question how these findings translate to the more complex natural language and mathematical reasoning tasks where pause tokens have been shown to help.

**Incomplete Separation for Log-Precision**: The paper does not prove a strict separation between logarithmic-precision Transformers with and without pause tokens (TF[L,L,0]⊊TF[L,L,P]), as this would require solving open problems in circuit complexity. While the authors are transparent about this limitation, it means the results for the log-precision case are less conclusive than for the constant-precision case.

---

> ### Author Rebuttal · Authors · 2025-07-31
>
> We thank the reviewer for their thoughtful comments and for recognising the novelty and clarity of our theoretical investigation into pause tokens.
>
> **On the gap between theory and practice:**
> Our main results establish equivalences between constant-depth Transformers with pause tokens and the well-studied circuit classes $\mathsf{AC}^0$ and $\mathsf{TC}^0$. These are general upper bounds that require a polynomial number of pause tokens to simulate arbitrary circuits from those classes. However, this does not preclude significantly smaller constructions for specific tasks. For example, if the task is evaluating DNF Boolean formula, the size of the circuit grows with the number of clauses, so if the number of clauses is constant, the number of pause tokens required will be constant. More generally, if the output only depends on a fraction of the input tokens (as in some math word problems or similar), the Transformer will not need a polynomial number of pause tokens in the entire input size, and may only need pause tokens corresponding to those positions. While the practical works cited (e.g. Goyal et al. 2024) observe benefits with a constant number of pause tokens, these do not aim for asymptotic circuit-level equivalence, and our results do not contradict them. In fact, as they have an upper bound on the input length, the number of pause tokens required is theoretically constant. That said, our results do imply that some dependence on input length is necessary in the worst case, given the strict separations without pause tokens.
>
> **On empirical scope:**
> We agree that broader empirical evaluation is a promising direction, but note that our contribution is primarily theoretical and motivated by existing empirical work. Several prior papers (including Goyal et al., 2024, and Pfau et al., 2024) have already demonstrated that pause tokens can improve performance on natural language and mathematical tasks, and our work aims to explain why this might be the case. We use the parity function in our tests due to previous works empirically demonstrating the difficulty in learning it without pause tokens, and because we can exactly characterize its circuit complexity (in $\mathsf{TC}^0$, not in $\mathsf{AC}^0$). This allows us to make the theoretical separation more empirically concrete. Extending these results to richer domains is a natural direction for follow-up work.
>
> **On the log-precision separation:**
> We appreciate the reviewer’s understanding on this point. As noted in the paper, separating TF[L,L,0] from TF[L,L,P] in the logarithmic-precision setting would require resolving major open problems in circuit complexity. It has been conjectured in some previous papers (see Merrill et al., 2023 and Pfau et al., 2024) that there exists a size hierarchy for $\mathsf{TC}^0$, which would suffice to prove a separation, but this conjecture remains open. While we cannot resolve this, we believe the formulation of the question in our framework is itself a meaningful step forward, and hope it invites further theoretical progress.

---

> > ### Author Response · Authors · 2025-08-07
> >
> > Dear Reviewer,
> >
> > We sincerely appreciate your feedback and have done our best to properly address all your comments. Do let us know if you have any additional questions. We would be very grateful for your consideration in updating our score, thank you!

---

### Official Review · Reviewer_KjRB · 2025-07-02

**Clarity:** 2
**Significance:** 3
**Originality:** 4
**Rating:** 4
**Confidence:** 3

**Summary:**

Attention mechanisms and transformer architectures have achieved significant advancements in natural language processing and computer vision, forming the foundation of many recent large-scale models. Incorporating intermediate reasoning steps and chain-of-thought prompting has led to improved performance across tasks such as mathematical reasoning and question answering. Recent studies have found that inserting intermediate, seemingly meaningless tokens—pause tokens—can also improve model performance, though this effect remains unclear within the COT framework. This paper provides a theoretical explanation for the benefit of pause tokens in enhancing transformer expressivity, grounded in circuit complexity. Specifically, it proves that 1) constant-precision transformers with a polynomial number of pause tokens match the expressivity of AC0 (constant-depth circuits with and/or/not gates), and 2) logarithmic-precision transformers with pause tokens correspond to TC0 (AC0 with threshold gates). These results support the conclusion that 3) pause tokens increase transformer expressivity. The theoretical claims are empirically validated on a parity benchmark, building on prior findings that parity is hard for transformers without pause tokens. Results show that pause tokens enhance performance on this task and enable causally masked transformers to learn globally dependent functions.

**Questions:**

Transformers have been used in many domains such as NLP and Computer vision. Could you please provide some tasks in various domains that can benefit from pause tokens while being tied to the proposed theories in this paper?


How does increasing the number of layers affect the number of pause tokens required and accuracy?

**Ethical Concerns:**

["NO or VERY MINOR ethics concerns only"]

**Final Justification:**

Thanks to the authors for their response to my comments. I still believe that adding more experiments on various datasets in different domains could better show the advantages of this study.

**Limitations:**

yes

**Quality:**

3

**Strengths And Weaknesses:**

Strength:
-- While transformers have achieved remarkable success across various domains, their theoretical foundations remain limited. This paper offers a theoretical analysis that clarifies the non-obvious reasons behind the performance gains observed when pause tokens are introduced.

-- Explaining the benefit of adding pause tokens to improve transformers’ expressivity through the lens of circuit complexity is both novel and non-trivial.

-- The results presented in Figure 1 and their analysis show the advantages of pause tokens on enhancing the transformer performance when the sequence length increases. This is particularly interesting because the transformers with pause tokens stay absolutely stable while other versions drop substantially. This shows the advantages of capturing global information once the pause tokens are introduced.

Weaknesses

-- The paper may be challenging to understand for a general audience due to its highly technical and theoretical nature. This could be improved by incorporating concrete examples, visualizations, and other intuitive elements to make the content more accessible and engaging for non-expert readers.

-- Although the abstract and introduction highlight the advantages of using pause tokens in QA and mathematical reasoning tasks, the experimental evaluation is limited to the parity benchmark. Expanding the evaluation to include additional datasets in QA and mathematical reasoning would strengthen the connection between the theoretical insights and real-world applications.

---

> ### Author Rebuttal · Authors · 2025-07-31
>
> We thank the reviewer for their time and thoughtful comments. We are glad they appreciated the theoretical contribution and the connection to Transformer expressivity.
>
> **On clarity and accessibility:**
> We appreciate the feedback and will add more concrete examples and intuitive explanations to improve accessibility. In particular, we will include examples of simple Boolean functions that are in $\mathsf{AC}^0$ and require different circuit sizes, and therefore different numbers of pause tokens n our framework.
>
> **On empirical scope:**
> We agree that broader evaluation is valuable. Our theoretical work was directly motivated by existing empirical results, particularly Goyal et al. (2023) and Pfau et al. (2024), which show that pause tokens improve performance on question answering and mathematical reasoning tasks. Our goal was to explain why this works, rather than to re-establish those results. To evaluate the alignment between theory and practice, we focused on the parity task, where the underlying complexity class (in $\mathsf{TC}^0$ but not in $\mathsf{AC}^0$), and the required $\mathsf{TC}^0$ circuit size, is precisely known. This allows us to make a concrete connection between circuit complexity and Transformer expressivity. We will highlight these connections and refer more explicitly to prior empirical results in the final version.
>
> **Question 1:**
> We will elaborate further in the final version, but to briefly address the question: while a direct mapping from $\mathsf{AC}^0$/$\mathsf{TC}^0$ to real-world NLP or vision tasks is difficult, prior work provides concrete empirical domains where pause tokens help. For example, Goyal et al. report gains in several mathematical and QA benchmarks, including GSM8K and CommonSenseQA, and Pfau et al. demonstrate improvements in multi-hop reasoning. These tasks involve long-range dependencies and intermediate computations, which are well modelled by the circuit-style reasoning our theory supports. We could also add some examples from formal language theory that may make the connection to NLP more concrete if you think it would be helpful.
>
> **Question 2:**
> This is addressed in Lemma 4.4 of the paper. We show that for any depth $l$, there exists $m > l$ such that $\mathsf{TF}_l[1,L,P] \subsetneq \mathsf{TF}_m[1,L,P]$. That is, increasing the number of layers strictly increases the class of functions the model can compute, even when pause tokens are available. This shows that depth and pause tokens are complementary mechanisms of expressivity. By Hastad’s switching lemma (1986), there also exist functions that require smaller circuits, and therefore fewer pause tokens, if the depth is increased. You may also find the concurrent work of Merrill et al. (2025) interesting, as they consider how increasing and even looping over depth affects expressivity in the presence of pause tokens (but only in the log-precision setting).

---

### Decision · Program_Chairs · 2025-09-17

**Decision:**

Accept (poster)

**Comment:**

**Summary of Scientific Claims and Findings**

The paper provides the first theoretical proof that adding "pause tokens" to constant-depth Transformers strictly increases their computational power. It formally connects constant-precision models with pause tokens to the complexity class $AC^0$ and logarithmic-precision models to $TC^0$. Empirically, it shows that pause tokens enable a Transformer to learn the parity function, a task it otherwise fails.


**Strengths**

* **Novelty:** It is the first paper to provide a rigorous theoretical explanation for the benefits of pause tokens.
* **Technical Soundness:** The theoretical results, grounded in circuit complexity, are considered correct and significant.
* **Clarity:** The paper is well-written and clearly articulates its contributions and the problem's motivation.


**Weaknesses of the Paper**

* **Theory-Practice Gap:** The theory requires a polynomial number of pause tokens, while practice often uses a small constant.
* **Limited Empirics:** Experiments are confined to the synthetic parity task, not the real-world tasks mentioned in the motivation.

**Reasons for Recommendation**

The paper makes a fundamental and novel contribution to our theoretical understanding of Transformers. It provides a solid foundation for an empirically observed phenomenon. The strengths of this foundational work outweigh the weaknesses, which are reasonably acknowledged as being outside the paper's primary scope. Thus I recommend acceptance.

**Summary of Discussion and Rebuttal**

The authors provided a strong rebuttal that successfully addressed reviewers' concerns.
* **On empirical scope and theory-practice gap:** The authors clarified their goal was theoretical explanation, not replication, and the polynomial token requirement is a worst-case bound. This satisfied reviewer **awq1**, who raised their score.
* **On mechanisms and comparisons:** They provided analysis of attention patterns and a clear distinction between single forward pass for pause tokens and multiple forward pass for Chain-of-Thought (CoT).
* **On technical clarifications:** They provided a concrete example (the Sipser function) for the theoretical separation when requested by **MSH9**.